# Study on spray combustion characteristics of liquid ammonia/dimethyl ether dual fuel based on different injection strategies

**Jun Fu**[1,2,3]*, **Han He**[1], **Feibin Yan**[4], **Yi Ma**[1], **Shuhui Wang**[1]

**1** College of Mechanical and Energy Engineering, Shaoyang University, Shaoyang, China, **2** Key Laboratory of Hunan Province for Efficient Power System and Intelligent Manufacturing, Shaoyang University, Shaoyang, China, **3** Key Laboratory of Hunan Province for Advanced Agricultural Machinery Equipment and Key Manufacturing Technology, Shaoyang University, Shaoyang, China, **4** College of Mechanical and Energy Engineering, Beijing University of Technology, Beijing, China

* 4160@hnsyu.edu.cn

## Abstract

Ammonia is a green zero-carbon fuel, yet its low reactivity poses challenges, including difficult ignition and slow combustion rates. Compared to diesel or biodiesel, dimethyl ether (DME) has no C-C bonds, which means it produces almost no soot and has a high cetane number that helps it ignite easily. So, using the highly reactive DME to help ignite liquid ammonia is a good way to make it burn better. This study uses computer simulations to look at how well liquid ammonia and DME work together as fuel with different injection setups. Results indicate optimal DME ignition enhancement at injector spacing $L = 6$ cm, injection angle 180°, and ammonia energy share 70%, outperforming cases with spacings of 7–8 cm and angles of 150°, 120°, 90°, and 60°. At the same time, having shorter spacing during DME combustion leads to smaller areas for OH but more $NH_2$ formation, showing that ammonia is more effective at cooling the flame and that more ammonia is being used as fuel. Additionally, when DME is not burning, both OH and $NH_2$ areas grow larger at shorter spacings, showing that the fuel mixes sooner, the reaction areas get bigger, and the burning process is more complete. Regarding the bimodal $NH_2$ peaks, the initial peak reflects partial ammonia oxidation that is flame-entrained during DME combustion, while the secondary peak indicates the onset of autoignition, which is characterized by diminished reaction rates and reduced combustion intensity.

## 1. Introduction

Ammonia ($NH_3$) as a carbon-free fuel [1], possesses significant advantages such as high energy density, ease of storage, and transportability, positioning it as a key alternative fuel for internal combustion engine systems [2]. Its key benefits include a freezing point of −33°C, making it easy to store and transport; it has more than

**Data availability statement:** All relevant data are within the manuscript and its Supporting information files.

**Funding:** This research was supported by the China Postdoctoral Science Foundation (Grant No. 2024M760170), the Hunan Provincial Natural Science Foundation of China (Grant No. 2023JJ50262), and the Hunan Provincial Department-level Project (Grant No. LXBZZ2024391). 1. Han He: Conceptualization, Data curation, Funding acquisition, Investigation, Resources, Writing – original draft. 2. Feibin Yan: Funding acquisition, Investigation, Methodology, Resources, Supervision. 3. Yi Ma:Funding acquisition, Resources, Validation, Visualization.

**Competing interests:** No authors have competing interests Enter: The authors have declared that no competing interests exist.

1.5 times the energy per volume compared to liquid hydrogen; it is cost-effective; and it is readily available, indicating good potential for industrial use. Owing to these advantages, ammonia is recognized as one of the most promising novel carbon-free alternative fuels [3–5]. As concluded by MacFarlane et al. [6], ammonia stands as one of the simplest electrofuels with the potential to become a primary vector for transportable renewable energy, displacing fossil fuels in applications such as power generation and transportation. Nevertheless, its practical implementation is hindered by inherent combustion limitations: low chemical reactivity, reduced heating value, slow flame speed (approximately one-fifth of methane's laminar flame speed) [7], and elevated autoignition temperature [8]. These properties collectively contribute to difficult ignition and incomplete combustion [9].

Research indicates [10] that pure ammonia engines require compression ratios exceeding 35:1. Under these conditions, heavy-duty diesel engines operating on pure ammonia necessitate approximately a 27:1 compression ratio and an autoignition temperature of 924 K to achieve sustained operation. The slow flame propagation speed makes pure ammonia generally unsuitable for practical engine operating requirements. However, Mounaïm et al. [11]found that engine operation failed due to insufficient compression ratio and the low flame speed of ammonia. This issue could be mitigated by introducing 10% hydrogen as a combustion promoter, indicating that ignition enhancers can effectively improve the ignition characteristics of liquid ammonia. Therefore, dual-fuel combustion strategies blending ammonia with highly reactive fuels are considered a promising solution.

Zhang et al. [12]conducted an experimental study on a high-pressure direct-injection (DI) engine operating with liquid ammonia/diesel dual-fuel and observed that the diesel flame accelerated the combustion of ammonia and enhanced the indicated thermal efficiency. Bjørgen et al. [13]employed an ammonia–diesel dual direct injection system in a single-cylinder engine to investigate the effects of injection strategies on engine performance and emissions. The results demonstrated that temporally overlapping the injection of diesel and ammonia achieved the best overall engine performance. Ebrahim et al. [14]investigated the combustion and emissions of a two-cylinder direct-injection engine operating on liquid ammonia/biodiesel dual fuel. Compared with pure biodiesel, the ammonia/biodiesel mixture elevated local in-cylinder temperatures and improved the indicated thermal efficiency. Shin et al. [15] studied how different ways of injecting ammonia/diesel blends affect their burning properties. The study showed that using ammonia/diesel blended fuel led to better combustion efficiency, less unburned ammonia ($NH_3$ slip), and lower NOx emissions compared to burning pure diesel. Nadimi et al. [16]investigated an ammonia–biodiesel dual direct injection engine through a combination of experimental and numerical approaches, with primary variables including ammonia energy fraction and ammonia injection timing. The results indicated that a higher ammonia energy fraction led to a stronger cooling effect and increased emissions of unburned ammonia.

Compared to pilot fuels like diesel or biodiesel, DME serves as a highly reactive e-fuel [17]. Due to its lack of C-C bonds, DME possesses a high cetane number and exhibits lower soot emissions [18,19]. Consequently, DME represents one of the

highly reactive, low-carbon fuels suitable for piloting ammonia combustion. Kong et al. [20]conducted an experimental study on a direct injection compression ignition (DICI) engine fuelled with an ammonia–DME blend. It was observed that reducing the DME proportion resulted in a shorter ignition delay, higher combustion temperatures, and reduced emissions of CO and HC. Zhang et al. [21] also found that adding DME greatly affects how ammonia burns at the beginning by creating $CH_3$ radicals. These radicals accelerated NO formation, causing NOx emissions to peak at an equivalence ratio of 0.9 and a DME mole fraction of 40%. Gross et al. [22] studied how liquid ammonia/DME burns in engines that mix fuel and air before ignition, and they found that using more DME leads to quicker ignition times. Tutak et al. [23] reported that DME positively impacts the combustion process, reducing ignition delay, shortening combustion duration, and effectively decreasing ammonia emissions. Ruiz-Gutiérrez et al. [24] studied how ammonia and DME ($NH_3$/DME) mixtures react when NO is added and when it is not. Their experiments showed that adding DME changes how ammonia burns by lowering the temperature needed for the reaction, and using more DME makes it harder for ammonia to fully convert during burning. Xiao et al. [25] investigated the laminar burning velocity of $NH_3$/DME/air mixtures under conditions of 0.1 MPa, 298 K, and equivalence ratios (φ) ranging from 0.7 to 1.5, using a constant-volume combustion chamber combined with experimental measurements and kinetic modeling. Their results demonstrate that co-firing $NH_3$ with dimethyl ether (DME) is an effective strategy to overcome the low reactivity of ammonia. In summary, DME serves as a low-carbon, high-reactivity pilot fuel that effectively enables liquid ammonia combustion and provides stable burning. However, mixing liquid ammonia and DME needs to be adjusted based on different engine conditions, like when the engine is running slowly with little load or quickly with a lot of load.

The dual-fuel direct injection strategy utilizing liquid ammonia and dimethyl ether (DME) can further enhance ammonia combustion and enable flexible adjustment of different injection strategies, making it one of the current key research priorities. Using high-pressure direct injection in the engine for liquid ammonia and DME dual fuel helps take full advantage of ammonia's zero-carbon emissions and DME's low-carbon, highly reactive benefits. Li B. et al. [26] showed that using high-pressure direct injection for dual fuel can lead to stable combustion even with 90% more air than needed. Adjusting the timing of DME injection can help manage nitrogen oxide ($NO_x$) emissions, but if the timing is set too early, it might cause the engine to misfire. Increasing the injection pressure makes the liquid ammonia spray break up better and improves the indicated thermal efficiency (ITE), but it also leads to higher $NO_x$ emissions. Compared to the standard engine, the improved high-pressure direct-injection dual-fuel (HPDF) mode using ammonia-DME shows a small drop in ITE but leads to big cuts in GHG and $NO_x$ emissions. However, there is not much research on the angle and distance between the liquid ammonia and DME injection jets, and we do not fully understand how these factors affect the way the spray burns.

In short, using a dual-fuel high-pressure direct injection (HPDF) method to burn liquid ammonia shows great promise for achieving clean and efficient combustion. This study uses a basic model for dual-fuel direct injection with detailed chemical reactions to create a combustion model that simulates how DME jets ignite liquid ammonia sprays. The model investigates how the included angle of ammonia/DME injection, the spatial configuration of the jet (including stand-off distance), and different energy fractions of liquid ammonia affect combustion. This framework aims to elucidate the fundamental ignition mechanisms governing ammonia/DME dual direct-injection combustion. The findings offer important lessons for optimizing injector arrangement design in liquid ammonia/DME dual-fuel direct-injection engines, thereby facilitating the practical implementation of ammonia fuel in power systems.

## 2. Theoretical model

### 2.1. KH-RT crushing model

Based on surface wave theory, the KH-RT breakup model gives a fairly accurate picture of how spray atomization works, allowing for dependable predictions of how far the spray goes and its shape. The KH spray model component, established through extensive experimental investigations by Reitz [27], computes droplet breakup primarily based on two key

parameters: the frequency of the maximum growing disturbance wave ($\mu_{KH}$)and its corresponding wavelength ($\mu_{KH}$). This study uses the KH-RT breakup model to describe how liquid ammonia and DME jets break apart when they hit a surface.

The expressions for the minimum droplet radius prior to breakup ($r_{KH}$), the characteristic breakup time ($r_{KH}$), and the velocity component of the newly formed droplet normal to the parent droplet's trajectory ($\nu_x$) are defined as follows:

$$r_{min} = B_x \lambda_{KH} \tag{1}$$

$$\tau_{KH} = \frac{3.762 B_\eta r_0}{\lambda_{KH} \mu_{KH}} \tag{2}$$

$$\nu_x = B_\mu \lambda_{KH} \mu_{KH} \tag{3}$$

Where $r_0$ denotes the radius of the parent droplet prior to breakup;

Similarly, the Rayleigh-Taylor (RT) breakup model computations are based on two analogous key parameters: the maximum growth rate ($\mu_{RT}$) and corresponding wavelength ($\lambda_{RT}$) of the dominant instability. The expressions for the minimum droplet radius prior to breakup ($r_{RT}$), characteristic breakup time ($\mu_{RT}$), and breakup length ($L_b$) are defined as follows:

$$r_{RT} = \frac{\lambda_{RT} C_\theta}{2} \tag{4}$$

$$\tau_{RT} = \frac{C_\omega}{\mu_{RT}} \tag{5}$$

$$L_b = C_\varsigma \sqrt{d_0 \frac{p_l}{p_g}} \tag{6}$$

Where $d_0$ denotes the injector orifice diameter, and $p_l$, $p_g$ represent the liquid and gas densities of the fuel droplet system, respectively, with units of $kg/m^3$.

In the above equations, $B_x$, $B_\eta$, $B_\mu$, $C_\theta$, $C_\omega$, $C_\varsigma$ represent the six breakup constants in the KH-RT breakup model. Due to differences in physical properties such as viscosity and density between liquid ammonia and DME, it is necessary to specify distinct spray model parameters for each fuel. These constants were calibrated based on the recommended values provided in Table 1, and the calibrated values are summarized in the same table.

**Table 1. Parameter settings for DME and liquid ammonia KH-RT models.**

| Project | Calibration value | | Recommended value |
|---|---|---|---|
| Fuel | Liquid ammonia | DME | – |
| $B_\eta$ | 8 | 11 | 5~100 |
| $B_x$ | 0.61 | | 0.61 |
| $B_\mu$ | 0.188 | | 0.188 |
| $C_\theta$ | 0.1 | | 0.1~1.0 |
| $C_\omega$ | 1 | | 0.1~1.0 |
| $C_\varsigma$ | 2 | | 0~50 |

## 2.2. Combustion model and turbulence model

This study primarily focuses on the DME-piloted ignition and combustion processes of liquid ammonia. To accurately simulate the highly complex physicochemical changes occurring during these processes, careful selection of a combustion model is essential. The SAGE presumed PDF model was employed to simulate the heat release from chemical reactions and molecular diffusion during fuel combustion. Based on chemical kinetic theory, this model enables the calculation of reaction rates for each elementary step in the reaction mechanism while solving the transport equations. Additionally, the RAG k–ε model, a Reynolds-Averaged Navier–Stokes (RANS) approach, was selected for turbulence modeling due to its favorable balance between computational accuracy and efficiency, making it well-suited for applied engineering studies.

Where $k_{fr}$ denotes the forward rate constant of the $r^{th}$ reaction, expressed in the Arrhenius form as follows [28]:

$$k_{fr} = A_r T^{b_r} e^{-\frac{E_r}{RT}} \tag{7}$$

The reverse rate constant $k_{br}$ for the $r^{th}$ reaction is related to the equilibrium constant $K_{cr}$ by the expression:

Where $A_r$ denotes the pre-exponential factor; T is the thermodynamic temperature; $b_r$ represents the temperature exponent; $E_r$ is the apparent activation energy; R is the universal gas constant, and $K_{cr}$ denotes the concentration-based equilibrium constant.

Based on the theory of chemical reaction kinetics, this model figures out the reaction speed for each basic step in the process while also solving the related transport equations. The $NH_3$/DME combustion mechanism chosen from Reference [25] includes 102 different substances and 594 basic reactions. This mechanism closely matches experimental results for how fast flames spread and how quickly they ignite with different $NH_3$/DME blend ratios, showing that it is a good choice for understanding how $NH_3$/DME mixtures burn.

## 3. Simulation model development and validation

### 3.1. Liquid ammonia spray

The numerical model for liquid ammonia spray was compared to experimental data from tests in a constant-volume chamber under inert conditions, as shown in Reference [29]. A numerical simulation model was established based on the experimental boundary conditions, with Table 2 detailing the simulation operating conditions. The RNG k-ε turbulence model was employed, with the constants governing compression and expansion modified to [30]. The computational domain comprised a 48 mm × 48 mm × 97 mm rectangular prism. A 0.22 mm injector orifice was positioned centrally on the top surface. To eliminate grid dependency in spray and combustion simulations, a grid independence study was conducted. Base grid sizes of 2 mm, 4 mm, and 6 mm were evaluated. In the area where the spray develops, fixed embedding was used along with 4 levels of adaptive mesh refinement (AMR) that adjusted based on changes in speed and temperature. Fig 1 demonstrates that simulation results using the 4 mm base grid exhibit negligible grid dependence. Therefore, the 4 mm base grid was adopted throughout this study to balance computational efficiency with solution accuracy. Fig 2 compares the simulation results with both the numerical and experimental spray morphologies obtained by Li et al. [29], in which the solid black line denotes the experimental liquid-phase ammonia spray, while the dashed black line represents the experimental vapor-phase ammonia spray. The results demonstrate that the spray model accurately predicts the morphology of liquid ammonia spray, validating its applicability in numerical simulations.

**Table 2. Simulation conditions of liquid ammonia spray.**

| Parameter | Injection Pressure/*MPa* | Ambient Temperature/*K* | Fuel Temperature/*K* | Environmental Gas | Environmental Density /*kg·m-3* |
|---|---|---|---|---|---|
| Numerical | 60 | 900 | 350 | $N_2$ | 18 |

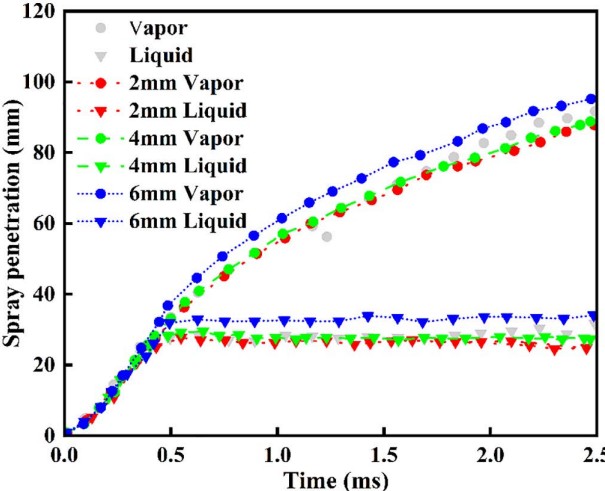

**Fig 1. Gas-phase and liquid-phase ammonia spray penetration distances.**

### 3.2. DME spray

DME spray simulations were carried out using experiments in a constant-volume chamber with both inert and reactive conditions, as mentioned in References [31,32]. Table 3 shows the conditions for the simulations, with the main case set at an injection pressure of 75 MPa, an ambient temperature of 900 K, an ambient density of 14.8 kg/m³, and a fuel temperature of 383 K. To keep things consistent with later simulations, the same grid size, refinement levels, and turbulence model settings used in the liquid ammonia model were applied to the DME simulations, using a 0.18 mm diameter for the injector orifice.

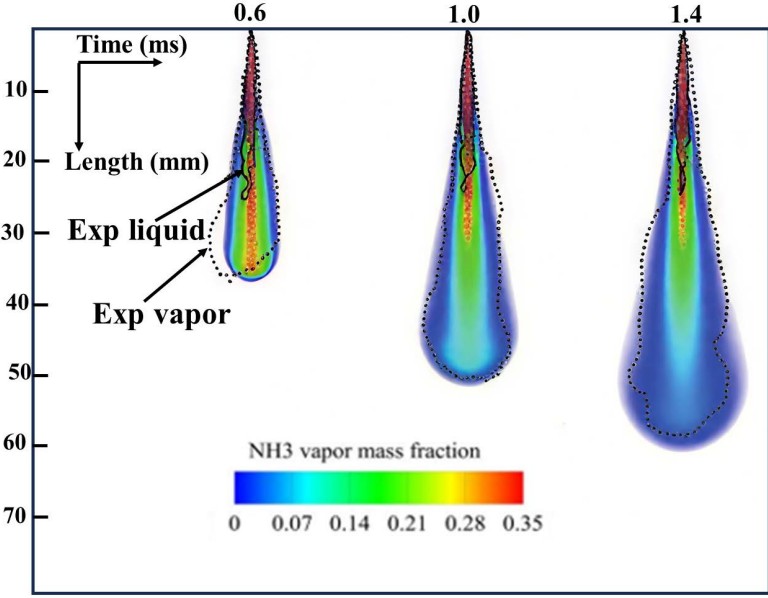

**Fig 2. Simulation and test result comparison of liquid ammonia spray morphologies.**

**Table 3. Simulation conditions of DME spray.**

| Parameter | Injection Pressure /MPa | Ambient Temperature/K | Fuel Temperature/K | Oxygen Volume Fraction/% | Environmental Density/kg·m-3 |
|---|---|---|---|---|---|
| Numerical | 75 | 900 | 383 | 15、18、21 | 14.8 |

Fig 3 shows a comparison between the simulated and actual measurements of how far the vapor and liquid spray travel in the DME baseline case when there is no reaction. While the simulated liquid penetration slightly exceeds experimental measurements in the later spray stages, the overall agreement remains satisfactory. The model demonstrates sufficient capability for predicting DME non-reacting spray behavior.

Fig 4 The figure compares simulated and experimental values of ignition delay period (ID) and flame lift-off length (LOL) under varying oxygen mole fractions. Here, ID is defined as the time interval from the start of injection (SOI) to the occurrence of the maximum temperature gradient. LOL represents the minimum axial distance where the Favre-averaged OH mass fraction reaches 14% of its maximum value [33]. The simulations show strong agreement with experimental data for both ID and LOL across different ambient densities. However, differences between the simulated and actual LOL values are seen when oxygen levels are low, mainly due to the shortcomings in the chemical kinetic mechanism. this study's focus on high-oxygen-concentration conditions, the DME spray model demonstrates high accuracy under both reacting and non-reacting conditions.

### 3.3. Numerical implementation scheme

The liquid ammonia model and DME model that have been verified to be accurate are simulated for high-pressure dual direct injection. Fig 5 shows the placement of the liquid ammonia and DME injectors to study the interaction between single-beam sprays. Table 4 shows Chamber specifications.The simulation conditions are shown in Table 5. The distance between the injectors is defined as L, the injection angle is defined as β, and the liquid ammonia energy ratio $\alpha_{NH_3}$ is defined as shown in Equation (8).

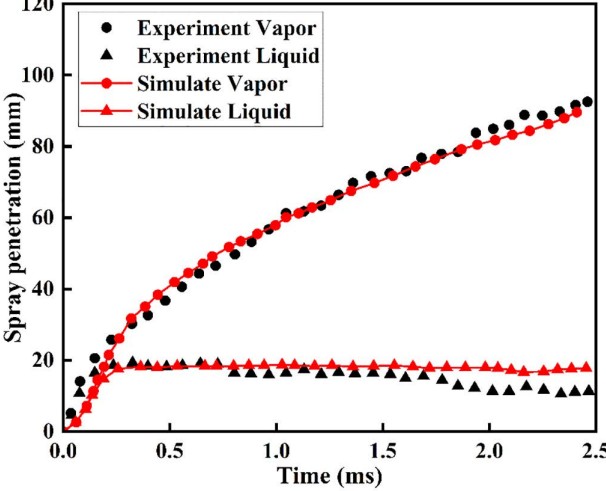

**Fig 3. Gas-phase and liquid-phase DME spray penetration distances under non-combustion conditions.**

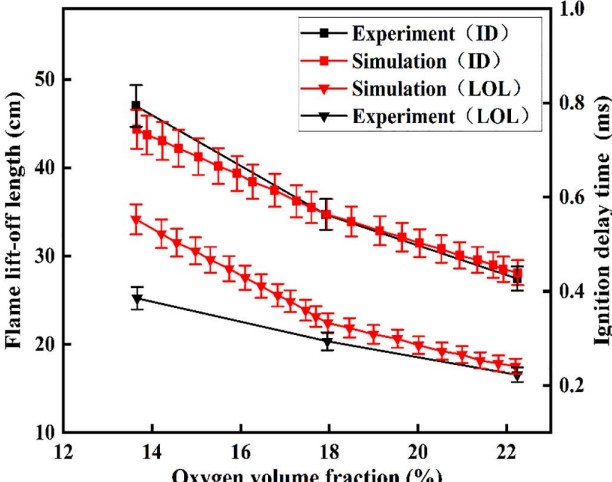

**Fig 4. Ignition delay period and flame lift-off lengths under different conditions.**

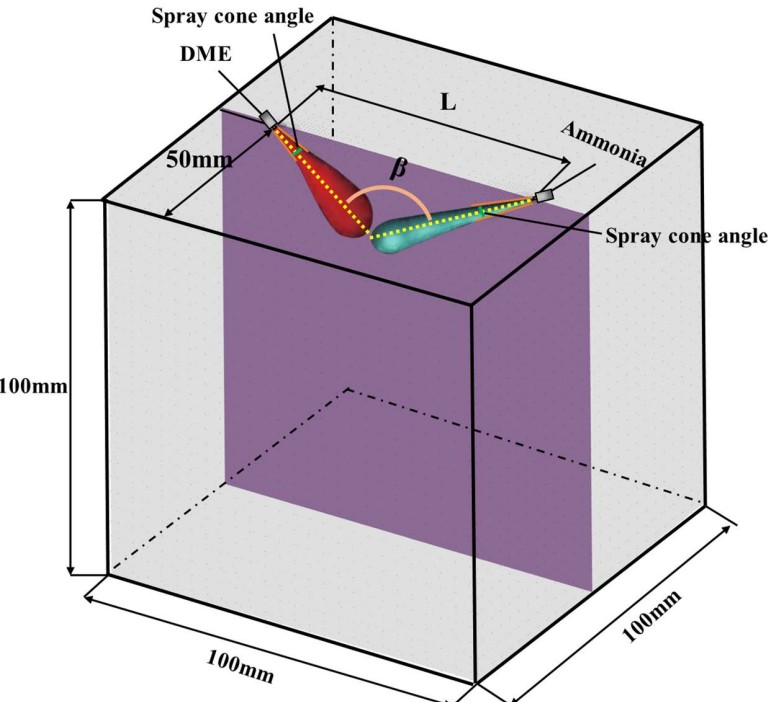

**Fig 5. Schematic diagram of geometric model of the constant volume bomb and fuel injector position.**

$$\alpha_{NH_3} = \frac{m_{NH_3}L_{NH_3}}{m_{NH_3}L_{NH_3} + m_{DME}L_{DME}} \tag{8}$$

Where: $m_{NH_3}$ = mass of liquid ammonia $(kg)$, $m_{DME}$ = mass of dimethyl ether $(kg)$, $L_{NH_3}$ = lower heating value (LHV) of liquid ammonia $(kJ/kg)$, $L_{DME}$ = lower heating value (LHV) of dimethyl ether $(kJ/kg)$.

**Table 4. Chamber specifications.**

| Parameter | Value |
| --- | --- |
| Chamber size/ mm | 100*100*100 |
| Ammonia nozzle diameter/mm | 0.22 |
| DME nozzle diameter/mm | 0.18 |
| Spray cone angle/° | 15° |

**Table 5. Simulation conditions.**

| Parameter | Injection Pressure /MPa | Injector distance /cm | Ambient Temperature/K | Ambient Density /kg·m-3 | Oxygen Volume Fraction/% | Injection Phase Difference/ms | Spray Angle /° | Liquid Ammonia Energy Fraction/ % |
| --- | --- | --- | --- | --- | --- | --- | --- | --- |
| Numerical | 75 | 6,7,8 | 900 | 14.8 | 23 | 0 | 180°, 150°, 120°, 90°, 60° | 70%, 80%, 90% |

Throughout this study, the total energy input was held constant. Variations in the ammonia energy fraction were achieved by modulating the injected masses of liquid ammonia and DME. The corresponding fuel masses for each ammonia energy fraction are documented in Table 6.

## 4. Results analysis and discussion

### 4.1. Analysis of ignition-combustion and pollutant formation characteristics with injector spacing in liquid ammonia-DME dual-fuel mode

The spray plume interaction dynamics governed by injector spacing critically modulates spray morphology evolution in liquid ammonia-DME dual-fuel systems, thereby exerting a dominant influence on flame development and pollutant formation pathways during the pilot-ignition phase. Fig 6 depicts the temperature distributions at ignition timing for opposed fuel injection ($\beta = 180°$) under varying injector spacings ($L = 6$ cm, $7$ cm, and $8$ cm). It is clearly observed that with $L = 6$ cm, liquid ammonia ignition occurs at t = 0.36 ms after DME injection. However, as the injector spacing increases from $L = 6$ cm to $L = 8$ cm, the initial contact between DME and liquid ammonia is progressively delayed, resulting in deteriorated pilot-ignition efficacy of ammonia. At $L = 7$ cm, the DME spray has not yet contacted the liquid ammonia spray at the ignition timing of 0.36 ms. In contrast, at $L = 8$ cm, the initial DME-ammonia contact occurs at 0.57 ms. Whereas at L = 6 cm, the liquid ammonia spray already interacts with DME at the 0.36 ms ignition timing, and the propagating pilot flame progressively ignites additional ammonia fuel as it develops.

The edge of the OH radical distribution is a good indicator of where the flame is, helping to describe how the flame moves during burning [34]. Figs 7 and 8 present the OH and $NH_2$ distributions for injector spacings of $L = 6$ cm, $7$ cm, and $8$ cm. As evidenced in Fig 7, at t = 0.36 ms after injection, only DME combustion occurs for $L = 7$ cm and L = 8 cm configurations. In contrast, at $L = 6$ cm, the liquid ammonia jet impinges on the DME flame front, inducing flame splitting into two distinct reaction zones. Subsequently, liquid ammonia jet impingement on the DME flame occurs at t = 0.57 ms for $L = 7$ cm and t = 0.75 ms for L = 8 cm. During the 0.57–0.9 ms period post-impingement, the OH zone area displays an

**Table 6. Fuel mass under different ammonia energy ratios.**

| Project | Parameter | | | |
| --- | --- | --- | --- | --- |
| Liquid Ammonia Energy Fraction/% | 0% | 70% | 80% | 90% |
| Liquid Ammonia Quality/mg | 0 | 15.9 mg | 18.2 mg | 20.4 mg |
| DME Quality/mg | 15 mg | 4.5 mg | 3 mg | 1.5 mg |

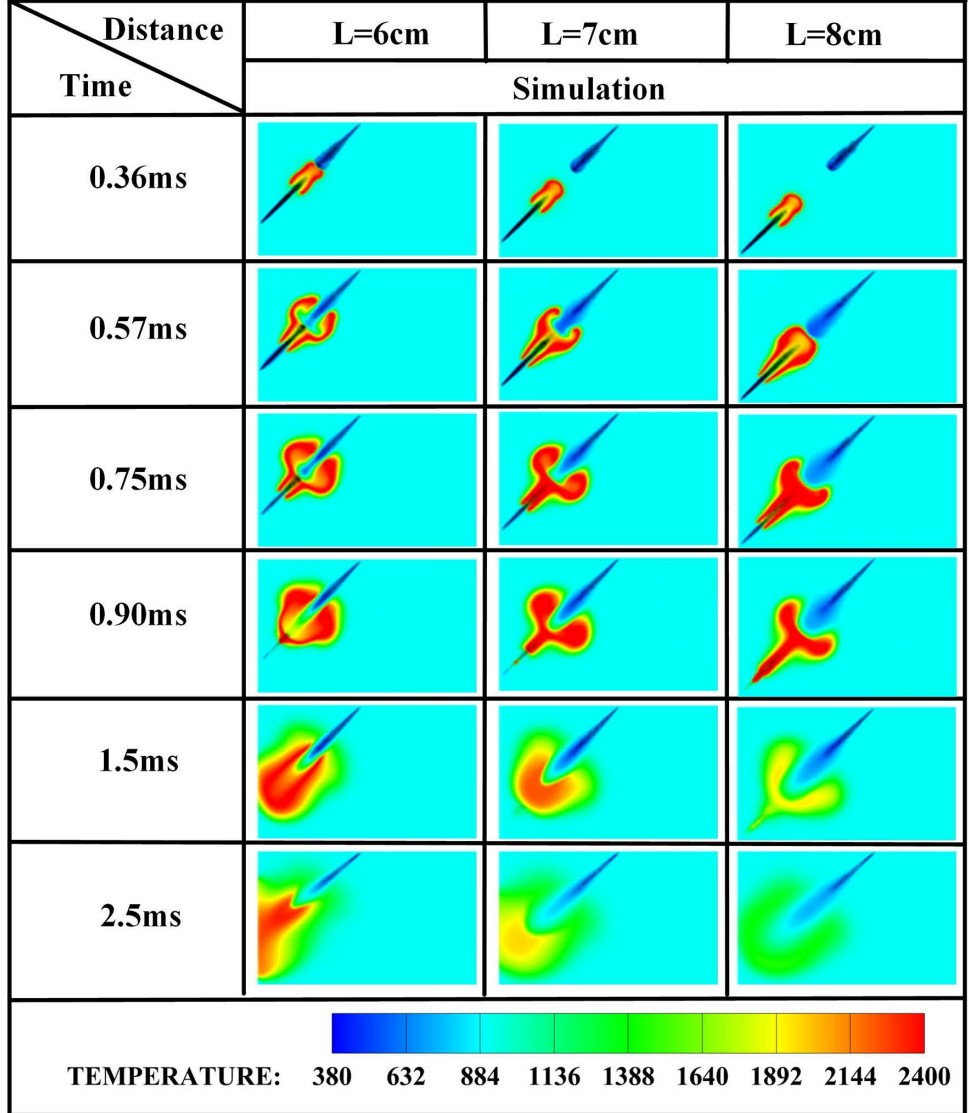

| Distance / Time | L=6cm | L=7cm | L=8cm |
|---|---|---|---|
| | Simulation | | |
| 0.36ms | | | |
| 0.57ms | | | |
| 0.75ms | | | |
| 0.90ms | | | |
| 1.5ms | | | |
| 2.5ms | | | |

TEMPERATURE: 380 632 884 1136 1388 1640 1892 2144 2400

**Fig 6. Temperature contours across injector spacings.**

inverse correlation with injector spacing, measuring largest at $L = 8$ cm, intermediate at $L = 7$ cm, and smallest at $L = 6$ cm. This trend indicates enhanced combustion promotion by ammonia jets at wider spacings, attributable to prolonged fuel-air mixing duration and increased heat absorption during spray evaporation. The emergence of $NH_2$ radicals signifies the onset of ammonia combustion [35]. Fig 8 demonstrates that the initial $NH_2$ appearance occurs at $t = 0.36$ ms, 0.51 ms, and 0.57 ms for injector spacings of $L = 6$ cm, 7 cm, and 8 cm, respectively. This temporal progression indicates a 0.15 ms/cm delay in $NH_2$ formation with increased spacing. Concurrently, combustion characteristics deteriorate progressively. During the identical post-impingement period ($t < 0.9$ ms), the $NH_2$ distribution area exhibits the largest magnitude at $L = 6$ cm, intermediate at $L = 7$ cm, and smallest at $L = 8$ cm. This pattern shows that having the injectors closer together helps the pilot ignition work better because it allows DME and ammonia to mix sooner and stay in contact longer, which means more ammonia is involved in the combustion. From $t = 0.9$ ms to 2.5 ms, both OH and $NH_2$ distribution areas increase

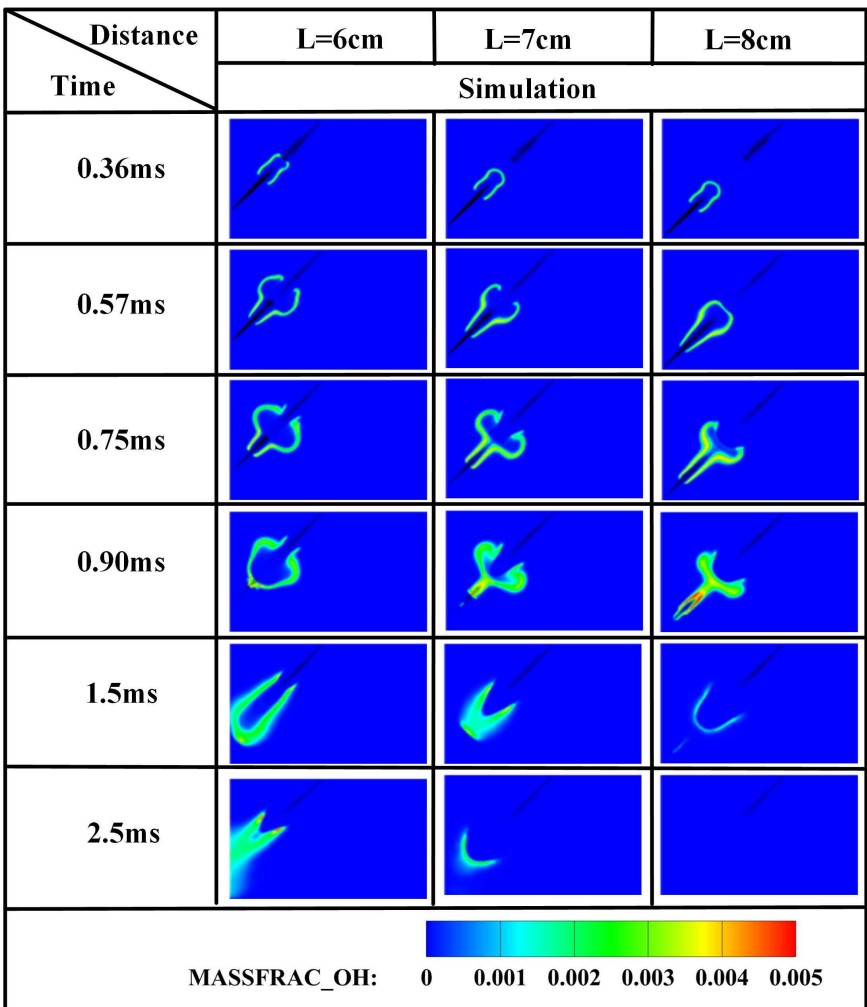

| Distance / Time | L=6cm | L=7cm | L=8cm |
|---|---|---|---|
| | Simulation | | |
| 0.36ms | | | |
| 0.57ms | | | |
| 0.75ms | | | |
| 0.90ms | | | |
| 1.5ms | | | |
| 2.5ms | | | |

MASSFRAC_OH:    0    0.001    0.002    0.003    0.004    0.005

**Fig 7. Spatial distribution of OH radicals.**

progressively as spacing decreases from $L = 8$ cm to $L = 6$ cm. This trend demonstrates progressively improved ignition performance with reduced spacing during this phase. The underlying mechanism relates to the fixed 0.7 ms DME injection duration: by $t = 0.9$ ms, when DME injection has essentially terminated, shorter initial spacing enables earlier fuel interaction, resulting in larger ammonia combustion areas. Fig 6 shows that there are larger areas of high temperature (over 1800 $K$) at a spacing of 6 cm compared to 7 cm and 8 cm, which confirms that closer injector spacing leads to better pilot ignition performance. As the spacing gets smaller, there is less OH distribution and more $NH_2$ formation during DME combustion, which shows that ammonia is more effective at cooling down the reaction and has a high energy requirement to turn into gas (1370 $kJ/kg$), while also allowing more ammonia fuel to participate. Additionally, the larger areas of OH and $NH_2$ during the ammonia burning stages (after DME is injected) show that combustion is more complete with shorter spacings, which is due to the fuel starting to interact sooner and leading to bigger reaction areas.

Concurrently, ammonia combustion generates NO and $NO_2$ emissions. NO formation predominantly localizes within high-temperature regions, exhibiting spatial correlation with OH radical distributions. The primary NO formation pathway in ammonia flames involves $NH_3$ decomposition into $NH_i$ intermediates followed by HNO oxidation [8]. There is a clear link

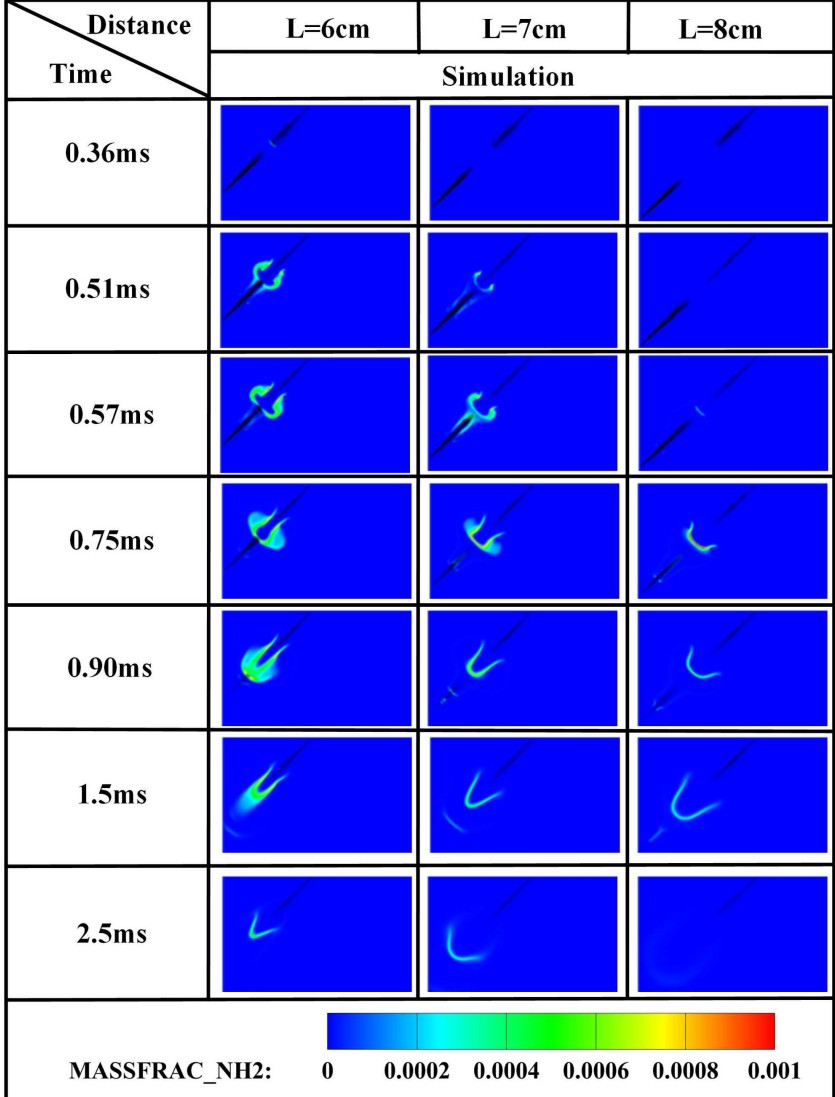

| Distance / Time | L=6cm | L=7cm | L=8cm |
|---|---|---|---|
| | Simulation | | |
| 0.36ms | | | |
| 0.51ms | | | |
| 0.57ms | | | |
| 0.75ms | | | |
| 0.90ms | | | |
| 1.5ms | | | |
| 2.5ms | | | |

MASSFRAC_NH2:   0   0.0002   0.0004   0.0006   0.0008   0.001

**Fig 8. Spatial distribution of NH$_2$ radicals.**

between the shapes of NH$_2$ and the OH profiles, showing that ammonia starts breaking down before the flame reaches its highest temperature. This pre-flame decomposition leads to NO formation along flame fronts, where fuel-NO mechanisms dominate in NH$_3$/DME flames—a phenomenon similarly documented in NH$_3$/H$_2$ flames [36]. Fig 9 shows that the first NO is formed at 0.36 ms when the spacing is 6 cm, which means there are ammonia-rich mixtures in the area where DME and ammonia interact in this setup. Comparatively, NO initiation delays occur at L = 7 cm and L = 8 cm spacings. Cross-referencing Figs 9 and 10 demonstrates significantly lower NO$_2$ production relative to NO, with NO$_2$ predominantly localized in intermediate-temperature regions (1000–1500$K$), particularly concentrated in mid-to-downstream sections of the DME spray plume.

Figs 11–13 present the combustion characteristics curves of OH, NH$_2$, and NOx, respectively, at various fuel injector distances. As evidenced in Fig 11, the sustained increase in OH concentration commencing at t = 0.36 ms signifies DME

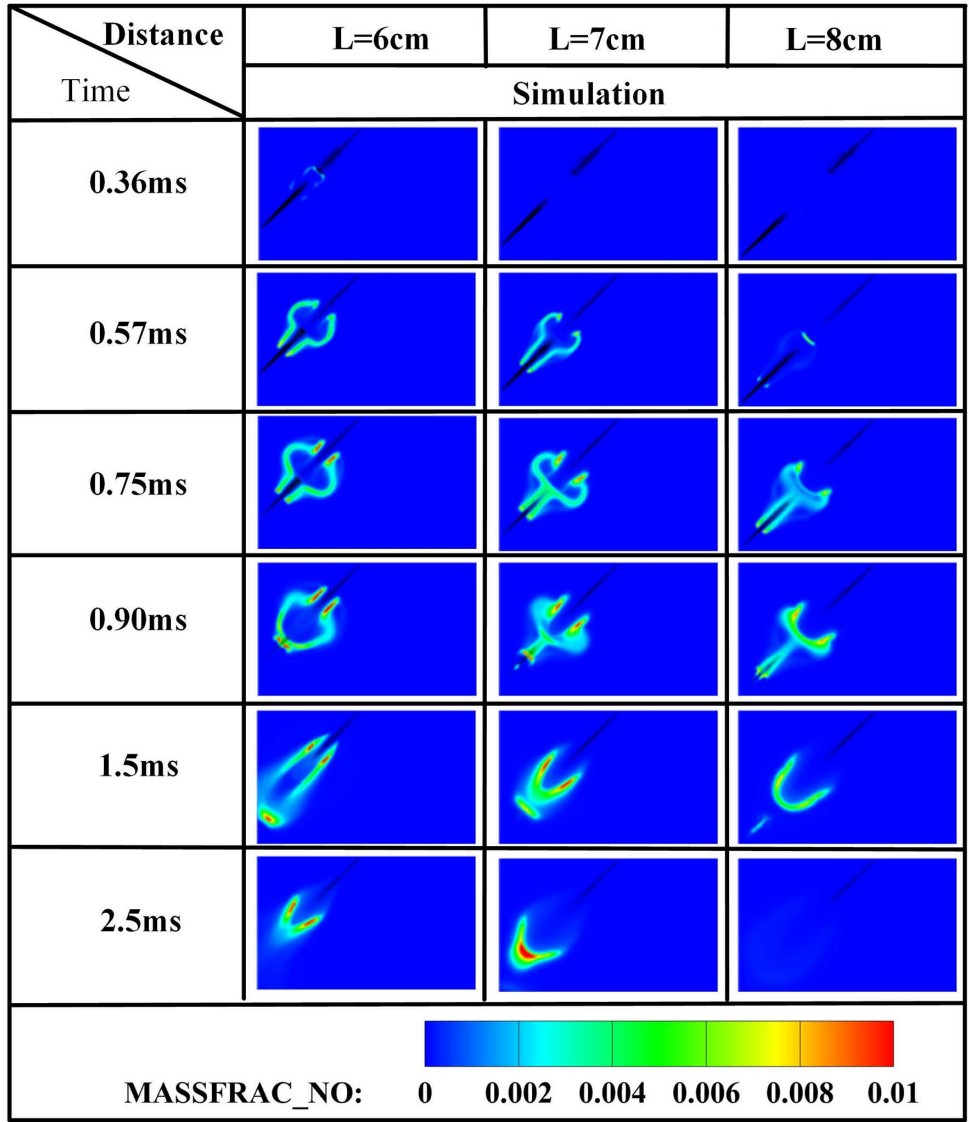

**Fig 9. Spatial distribution of NO radicals.**

ignition initiation and subsequent flame kernel development. Across all injector spacings, OH concentrations exhibit a characteristic rise-decline profile, with peak magnitudes increasing as spacing decreases. This enhancement stems from earlier DME-ammonia contact initiation (advancing by 0.15 ms/cm spacing reduction) and extended interaction duration, promoting greater ammonia entrainment into the reaction zone. During the fixed 0.7 ms DME injection period prior to $t = 0.7$ ms, OH formation rates maintain comparable temporal trends regardless of spacing variations. Both $L = 7$ cm and $L = 8$ cm spacings exhibit an identical sequence: DME autoignition precedes OH formation rate peaking. Subsequently, inflection points emerge in OH decay rates around $t = 1.5$ ms, where decline gradients attenuate. Cross-referencing Fig 7 contours confirms this transition corresponds to ammonia combustion onset, augmenting OH replenishment. Crucially, ammonia ignition initiates only after peak DME combustion intensity at these wider spacings, with $L = 7$ cm demonstrating superior ignition efficacy versus $L = 8$ cm. This delayed ignition relative to optimal DME thermal conditions directly

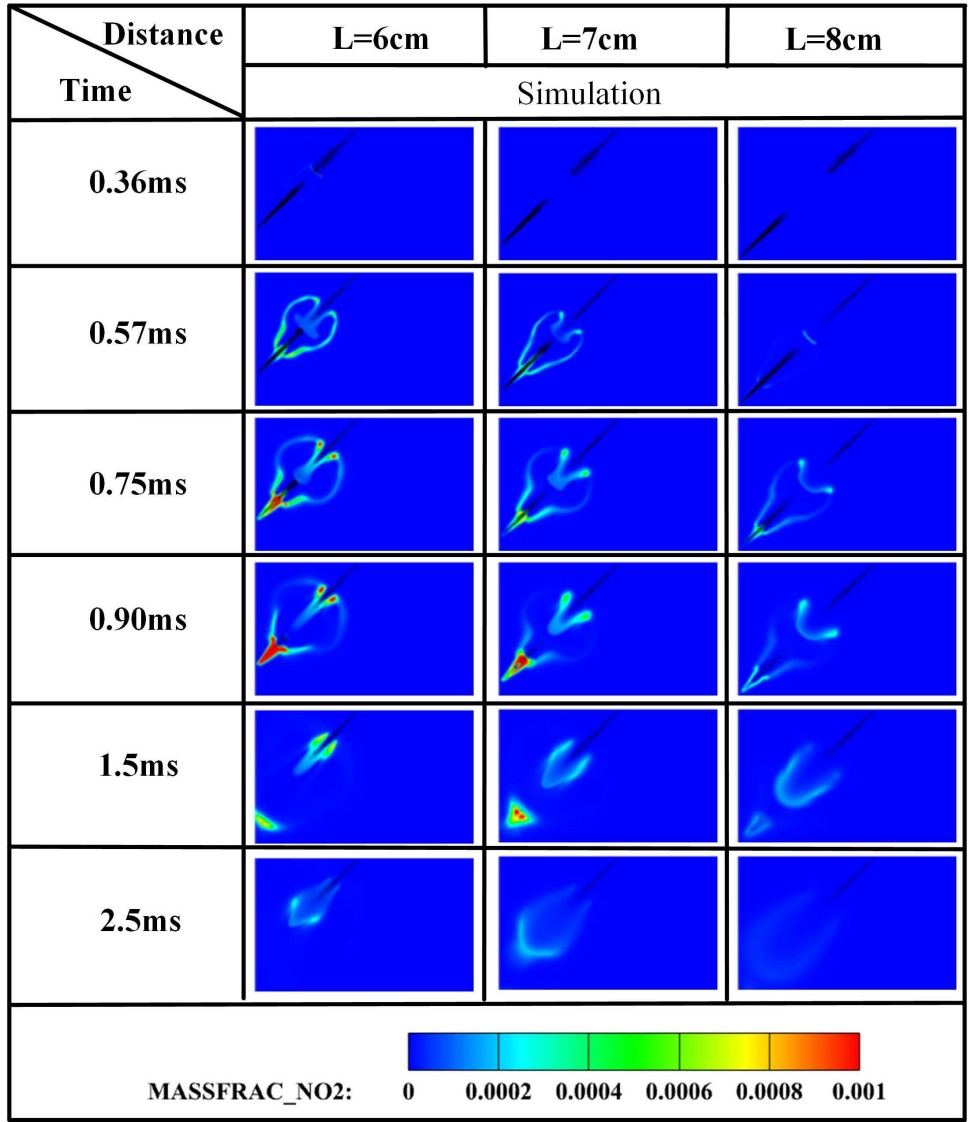

| Distance / Time | L=6cm | L=7cm | L=8cm |
|---|---|---|---|
| | Simulation | | |
| 0.36ms | | | |
| 0.57ms | | | |
| 0.75ms | | | |
| 0.90ms | | | |
| 1.5ms | | | |
| 2.5ms | | | |

MASSFRAC_NO2:  0  0.0002  0.0004  0.0006  0.0008  0.001

**Fig 10. Spatial distribution of NO₂ radicals.**

compromises ammonia ignition reliability, establishing that excessive injector spacing detrimentally impacts flame propagation dynamics. However, increased injector spacing extends the fuel-air mixing preparation period for liquid ammonia sprays. At $t = 0.7$ ms, OH production rates at $L = 7$ cm and $L = 8$ cm exceed those at $L = 6$ cm, indicating that wider spacings enhance atomization quality and promote OH radical generation. Conversely, during $t = 1.1$-$1.5$ ms for $L = 6$ cm spacing, the OH production rate stabilizes—a phase-locked combustion transition where DME pilot flames ignite ammonia while sustaining OH formation. Subsequently, ignited ammonia undergoes rapid combustion, sharply increasing OH production until fuel depletion eventually reduces OH generation rates. This confirms that at $L = 6$ cm spacing, ammonia ignition occurs concurrently with peak DME combustion intensity, thereby accelerating rapid ammonia oxidation. Fig 11 demonstrates that reduced spacing advances ammonia ignition timing and elevates the highest OH production rates by 18-25%. Crucially, while the $L = 7$ cm and $L = 8$ cm configurations experience ammonia ignition after DME's OH production peak, the $L = 6$ cm

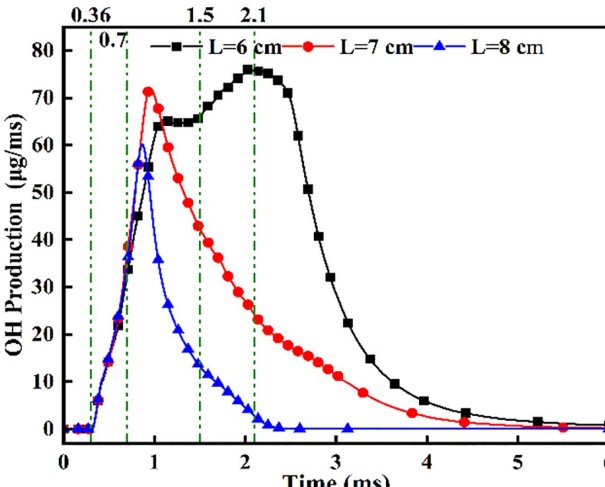

**Fig 11. OH Radical Evolution at Varied Inter-Injector Distances.**

case achieves synchronized ignition precisely at this peak. This temporal coincidence maximizes thermal energy transfer from DME combustion, enhancing ammonia combustion completeness through optimal flame temperature maintenance.

Fig 12 demonstrates that increased injector spacing reduces $NH_2$ formation rates by 15–22% while delaying the inflection timing by 0.12 ms/cm. As confirmed by $NH_2$ distribution contours in Fig 8, the earliest fuel interaction at $L = 6$ cm spacing initiates premature $NH_2$ concentration rise, culminating in peak combustion intensity at $t = 0.7$ ms. The $L = 7$ cm and $L = 8$ cm configurations exhibit intermediate behavior. All three spacings reach peak $NH_2$ concentrations at $t = 0.7$ ms, coinciding with DME injection termination and complete fuel interaction, thereby initiating sustained ammonia combustion. The $L = 6$ cm case demonstrates the earliest fuel contact, maximum $NH_2$ production, and optimal combustion efficiency. Regarding the post-0.7 ms $NH_2$ profile decline-resurgence pattern at $L = 7/8$ cm: post-injection temperature decay occurs due to extended DME-ammonia interaction time (0.19 ms/cm delay), causing insufficient thermal energy at wider spacings,

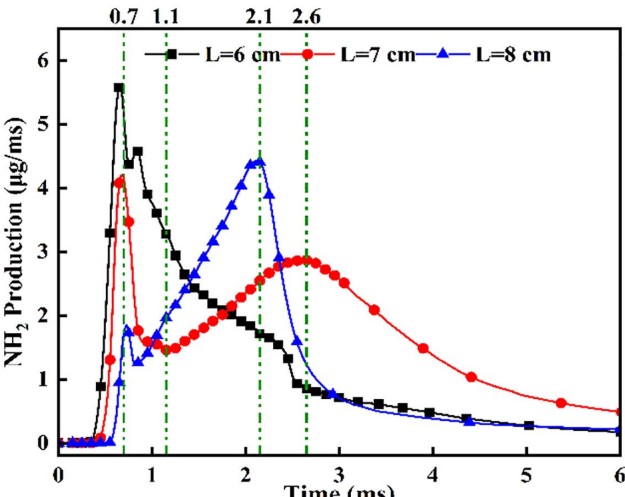

**Fig 12. $NH_2$ Formation Dynamics Across Injector Spacing Configurations.**

which reduces reaction intensity. Subsequent ammonia autoignition during $t=1.1$–$2.6$ ms then drives $NH_2$ resurgence. Specifically for $L=7$ cm, $NH_2$ concentrations decline after $t=0.7$ ms, followed by resurgence during $t=1.1$–$2.6$ ms as auto-ignition commences. For the $L=8$ cm configuration, increased injector spacing enhances fuel-air mixing quality, enabling the DME flame front to propagate through premixed ammonia envelopes. Consequently, post-0.7 ms $NH_2$ concentration exhibits an attenuated decline rate, followed by accelerated resurgence culminating in peak formation at $t=2.1$ ms. With constant total ammonia mass injected, the bimodal $NH_2$ profiles at $L=7$ cm and $L=8$ cm spacings demonstrate distinct combustion phases: the initial peak signifies partial ammonia entrainment into the DME flame front, reaching maximum intensity at 0.7 ms before declining, while the secondary peak indicates ammonia autoignition characterized by diminished reaction rates and combustion intensity, with post-2.6 ms decay confirming near-complete fuel depletion. This profile evolution reveals that reduced spacing universally produces initial $NH_2$ rate increase-decrease sequences. However, the resurgence-decline patterns exclusively observed at wider spacings ($L=7/8$ cm) result from the reignition of recirculated ammonia fuel, which occurs under suboptimal thermal conditions due to misalignment with peak DME combustion temperatures, ultimately yielding 18–25% lower cumulative $NH_2$ production compared to $L=6$ cm.

Fig 13 shows the NOx variation curves at different fuel injector distances. Fig 13 demonstrates NOx concentration ascent commencing at $t=0.36$ ms post-DME ignition, with distinct concentration gradients across spacings: steepest at $L=8$ cm, intermediate at $L=7$ cm, and gentlest at $L=6$ cm. This progression arises from enhanced ammonia atomization at wider spacings, where DME flame fronts initially ignite atomized ammonia layers. Oxygen-enriched conditions at $L=8$ cm accelerate NOx formation rates versus narrower spacings. Beyond $t=1.0$ ms, the $L=8$ cm profile plateaus, followed by $L=7$ cm at $t=2.0$ ms, while $L=6$ cm maintains ascent until $t=2.7$ ms, indicating prolonged ammonia combustion after pilot ignition at reduced spacings.

Typically, 5%(5% Cumulative heat release,CHR05), 50%(50% Cumulative heat release,CHR50), and 90% (90% Cumulative heat release,CHR90)of the cumulative heat release are used to denote the start of combustion, the center of combustion, and the end of combustion, respectively [37]. Figs 14 and 15 presents an analysis of the effects of different fuel injector distances on ignition combustion and pollutant formation. As shown in Fig 14, at an initial temperature of 900 K, the ignition delay period decreases gradually as the injector distance increases from $L=6$ cm to $L=8$ cm. This trend can be attributed to enhanced atomization of liquid ammonia at larger distances, which allows the flame front of dimethyl ether (DME) combustion to ignite the atomized ammonia layer more effectively. In contrast, the combustion duration initially

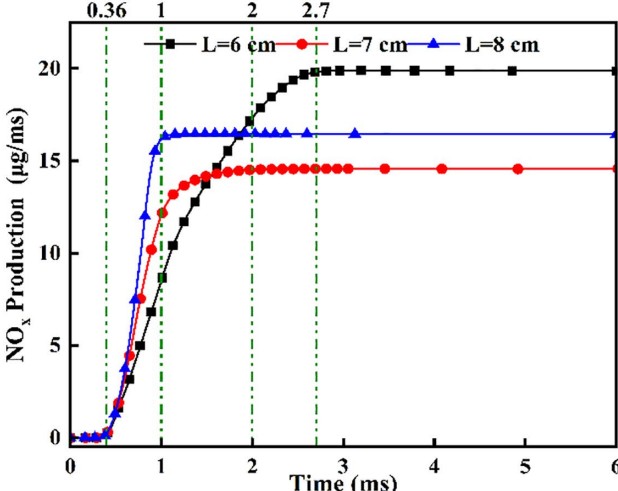

**Fig 13. NO$_x$ Emission Trends Versus Injector Separation Distance.**

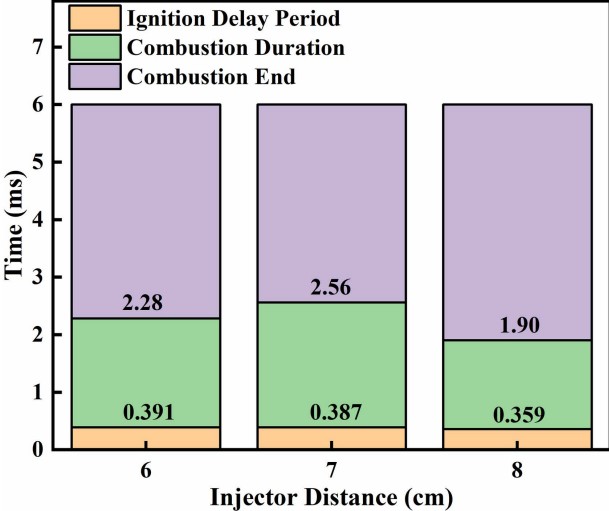

**Fig 14. Combustion phasing under different fuel injector distances.**

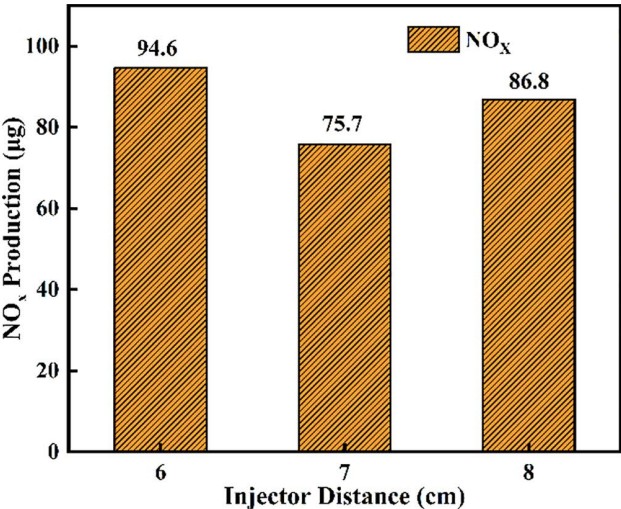

**Fig 15. NOx Production Rates as Functions of Nozzle Spacing.**

increases and then decreases with increasing injector distance. At L=6 cm, earlier contact between ammonia and DME results in heat absorption by the ammonia spray, leading to a shorter combustion duration compared to L=7 cm. Fig 15 shows the NOx production under different fuel injector distances. It can be clearly observed that the cumulative NOx emissions generated during the DME-ignited ammonia combustion process within 6 ms are the highest at L=6 cm, followed by L=8 cm, while the lowest NOx production occurs at L=7 cm. This trend is consistent with the NOx variation curves under different injector distances shown in Fig 13.

Under a fixed ammonia energy share ($\alpha$=80%) with an opposed dual-injector configuration ($\beta$=180°), Figs 16 and 17 presents maximum temperatures and heat release rates for DME-piloted ammonia combustion across injector spacings. Fig 16 reveals a rapid temperature rise to approximately 2700 $K$ commencing at $t$=0.36 ms post-DME ignition, sustained

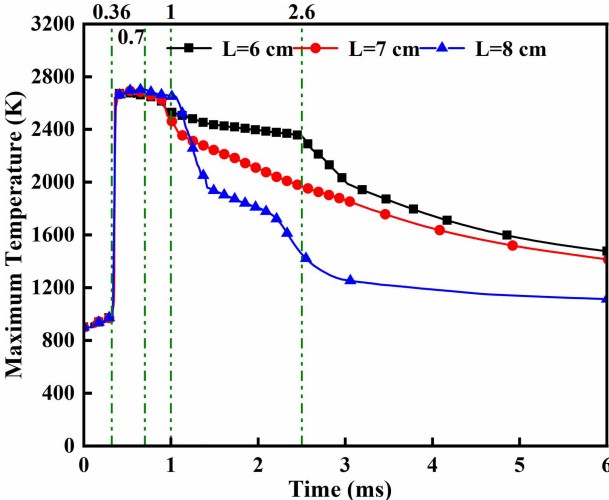

**Fig 16. Maximum temperature.**

through $t = 0.7$ ms due to continuous DME injection. This thermal plateau correlates with DME's 350°C ignition point and 0.7 ms injection duration. Subsequent bifurcated temperature decline beyond $t = 0.7$ ms stems from evaporative cooling during flame-ammonia interaction. Between $t = 0.7$ and 1.0 ms, $L = 6$ cm exhibits the most pronounced cooling rate, while $L = 8$ cm maintains marginally higher temperatures (by 80–120 $K$) than $L = 6/7$ cm, attributable to extended fuel interaction time enhancing atomization quality. Crucially, $L = 6$ cm sustains combustion temperatures above 2400 $K$ during $t = 1.0$–2.6 ms, demonstrating prolonged reaction stability. Fig 17 shows progressively reduced maximum heat release rates with increased spacing. Divergence in heat release profiles emerges after $t = 0.36$ ms, directly linked to spacing-dependent fuel contact timing. The accelerated rate rise at $L = 6$ cm during $t = 0.36$–1.0 ms results from earlier and more complete fuel mixing within optimal ignition temperature windows, whereas delayed contact at wider spacings compromises combustion efficiency.

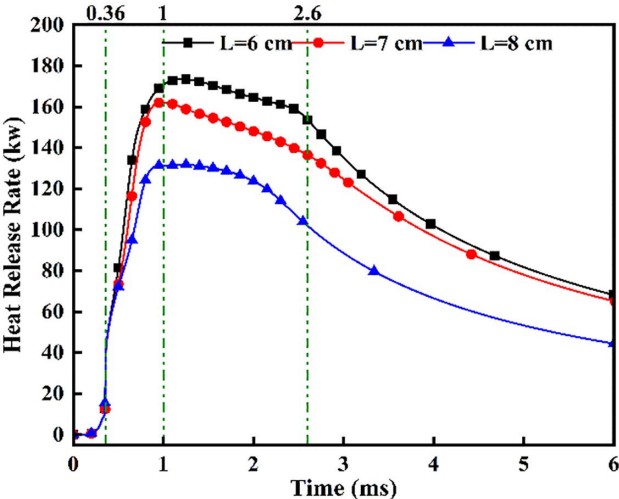

**Fig 17. Heat Release Rate.**

## 4.2. Analysis of ignition-combustion and pollutant formation characteristics with injection angle in liquid ammonia-DME dual-fuel mode

Injection angle constitutes a critical parameter governing DME-piloted ammonia ignition and pollutant formation. Building upon prior findings confirming optimal combustion efficiency at $L = 6$ cm with $\alpha = 80\%$, this study investigates five injection angles ($\beta = 180°$, 150°, 120°, 90°, 60°) while maintaining constant spacing and fuel mass. Fig 18 presents ignition-phase temperature contours under varied angles, revealing progressively extended fuel contact times with decreasing $\beta$: immediate contact at $\beta = 180°$ ($t = 0.36$ ms), imminent contact at $\beta = 150°$ ($t = 0.36$ ms), and measurable delays of 0.45 ms, 0.57 ms, and 0.90 ms for $\beta = 120°$, 90°, and 60°, respectively. Crucially, at $\beta = 180°$, ammonia-DME interaction commences synchronously with DME ignition, propelling sustained flame propagation as ammonia spray continuously feeds the reaction front. Comparative analysis confirms significantly larger high-temperature zones (>2000 $K$) post-$t = 0.9$ ms at $\beta = 180°$ versus narrower angles, demonstrating superior combustion efficacy.

Figs 19−22 depict OH, NH$_2$, NO, and NO$_2$ distributions for injection angles $\beta = 180°$, 150°, 120°, 90°, and 60°. Fig 19 reveals exclusive DME combustion at $\beta = 150°$-60° at $t = 0.36$ ms, whereas $\beta = 180°$ exhibits ammonia jet impingement on burning DME, inducing flame segmentation. Delayed ammonia-DME contact occurs at narrower angles versus $\beta = 180°$. During 0-0.9 ms, OH distribution areas decrease progressively with increasing $\beta$ (largest at $\beta = 60°$, smallest at $\beta = 180°$), reversing post-$t = 0.9$ ms with maximum OH at $\beta = 180°$. This inversion confirms enhanced combustion promotion at wider angles, attributable to earlier fuel interaction and optimized heat absorption dynamics. Cross-referencing Figs 19 and 20 demonstrates localized high-temperature zones at spray heads upon ammonia contact, producing NH$_2$ radicals that expand radially with time. NH$_2$ emergence delays progressively from $t = 0.36$ ms ($\beta = 180°/150°$) to 0.45/0.57/0.90 ms ($\beta = 120°/90°/60°$). Correspondingly, NH$_2$ distribution areas during 0-0.9 ms decrease sequentially from $\beta = 180°$ to 60°,

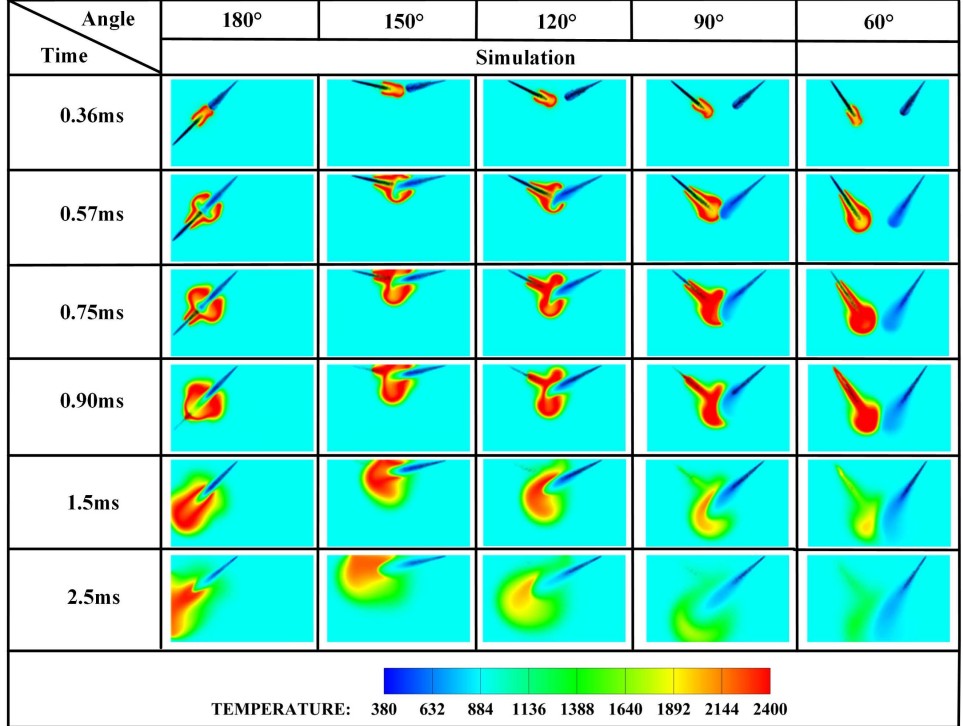

**Fig 18. Temperature contours across injector angles.**

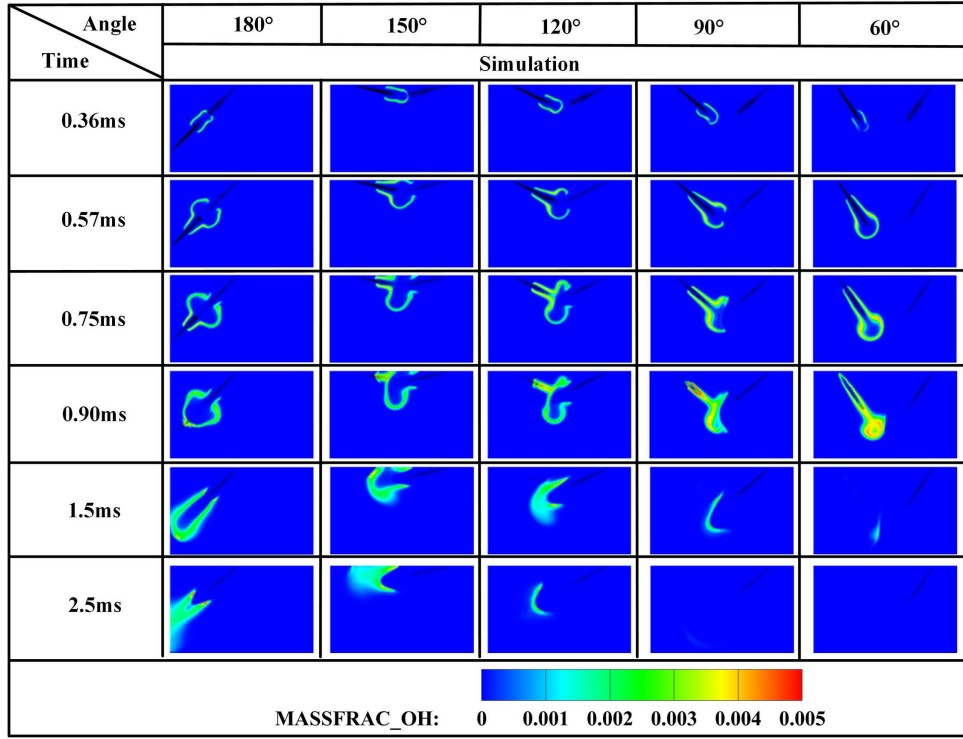

| Angle Time | 180° | 150° | 120° | 90° | 60° |
|---|---|---|---|---|---|
| | Simulation | | | | |
| 0.36ms | | | | | |
| 0.57ms | | | | | |
| 0.75ms | | | | | |
| 0.90ms | | | | | |
| 1.5ms | | | | | |
| 2.5ms | | | | | |

MASSFRAC_OH:    0    0.001    0.002    0.003    0.004    0.005

**Fig 19. Spatial distribution of OH radicals.**

evidencing superior pilot-ignition efficacy at wider angles due to prolonged interaction duration. Post-$t$ = 0.9 ms, concurrent expansion of OH and NH$_2$ zones occurs, with $\beta$ = 180°/150° achieving complete flame encapsulation and optimal combustion efficiency, whereas acute angles ($\beta$ ≤ 120°) yield incomplete ammonia oxidation due to delayed high-temperature exposure.

Figs 21 and 22 display NO and NO$_2$ distributions across injection angles, demonstrating rapid pollutant formation upon ammonia combustion initiation. Critically, both NO and NO$_2$ production decrease progressively with increasing injection angle ($\beta$ = 180° → 60°). This inverse correlation stems from delayed fuel interaction at wider angles, where ammonia misses the optimal DME ignition window, resulting in reduced combustion participation and incomplete oxidation that collectively suppress NOx formation by 25–40% compared to acute angles.

Figs 23−25 present the combustion characteristics curves of OH, NH$_2$, and NOx, respectively, under different injection angles. Fig 23 indicates OH concentration ascent commencing at $t$ = 0.36 ms, signifying DME ignition initiation. Near-identical OH evolution during $t$ = 0.36–0.7 ms reflects consistent DME combustion dynamics under the fixed 0.7 ms injection duration. Crucially, during $t$ = 0.7–1.2 ms, steeper OH concentration gradients emerge at $\beta$ = 120°, 90°, and 60° versus wider angles (180°/150°), attributable to enhanced atomization and prolonged fuel interaction facilitating ammonia entrainment into DME flame fronts. However, diminished combustion efficiency at acute angles subsequently reduces OH production beyond $t$ = 1.2 ms due to suboptimal ignition conditions. Conversely, sustained OH increase during $t$ = 1.1–2.1 ms at $\beta$ = 180° confirms supplementary OH generation from ammonia combustion. Notably, peak OH magnitude at $\beta$ = 150° demonstrates effective thermal support for ammonia combustion prior to $t$ = 1.2 ms, while post-1.2 ms decline indicates insufficient flame temperature maintenance for sustained ammonia ignition after spray evaporation cooling.

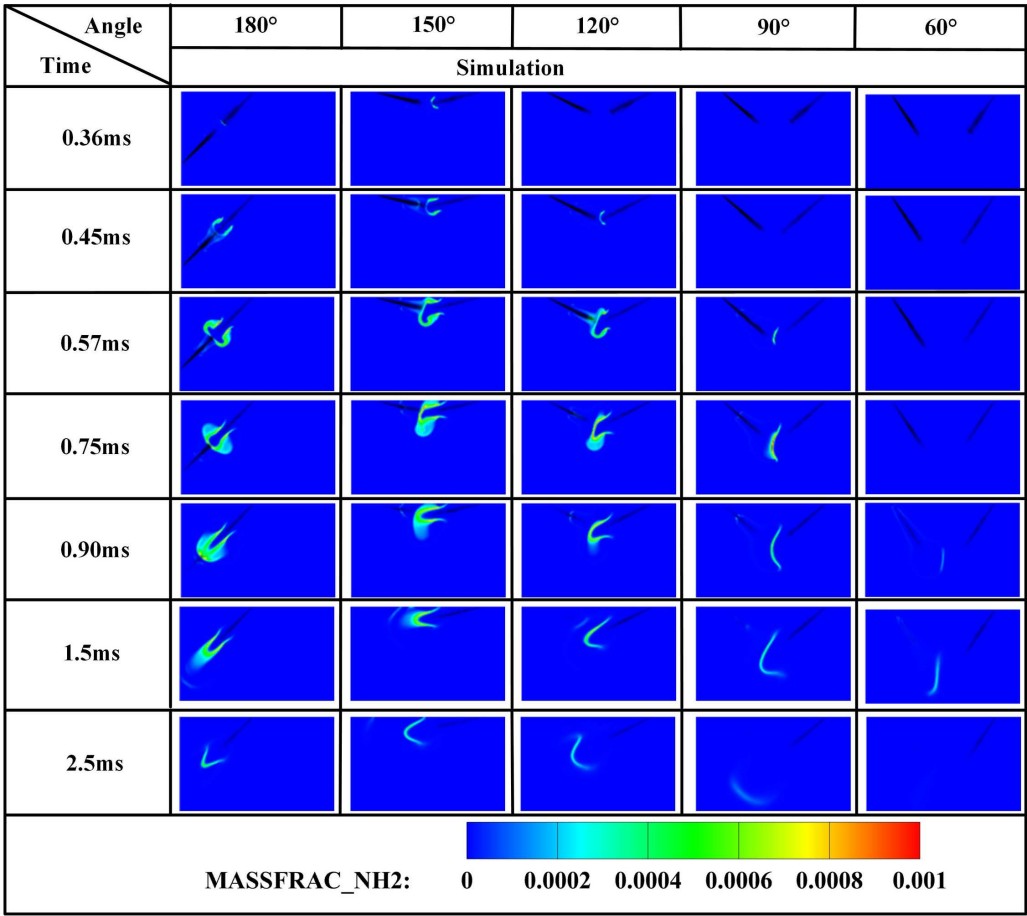

| Angle Time | 180° | 150° | 120° | 90° | 60° |
|---|---|---|---|---|---|
| | Simulation | | | | |
| 0.36ms | | | | | |
| 0.45ms | | | | | |
| 0.57ms | | | | | |
| 0.75ms | | | | | |
| 0.90ms | | | | | |
| 1.5ms | | | | | |
| 2.5ms | | | | | |

MASSFRAC_NH2:  0  0.0002  0.0004  0.0006  0.0008  0.001

**Fig 20. Spatial distribution of NH$_2$ radicals.**

Fig 24 demonstrates progressively reduced NH$_2$ peak production and formation rates with decreasing injection angles. Cross-referencing Fig 20 contours confirms earliest ammonia-DME contact at $\beta = 180°$, initiating premature NH$_2$ concentration rise culminating in peak combustion intensity at $t = 0.7$ ms. Conversely, NH$_2$ emergence delays occur at acute angles, though all configurations reach maximum concentrations near $t = 0.7$ ms due to synchronized fuel interaction upon DME injection termination. Maximum NH$_2$ yield occurs at $\beta = 180°$, followed by $\beta = 150°$, while $\beta = 120°/90°/60°$ exhibit post-0.7 ms decline-resurgence patterns. This secondary decline stems from extended DME-ammonia interaction time reducing subsequent combustion temperatures. Resurgence at acute angles reflects enhanced atomization enabling flame propagation through premixed ammonia envelopes. Crucially, $\beta = 180°/150°$ profiles display unimodal rise-decay sequences indicating direct transition from piloted to autoignited combustion, whereas $\beta = 120°/90°$ show bimodal characteristics: the first peak ($t \leq 0.7$ ms) signifies flame-entrained partial oxidation, and the secondary peak indicates delayed autoignition with diminished reaction intensity (38–45% lower than $\beta = 180°$). The $\beta = 60°$ profile confirms only vapor-phase ammonia ignition without sustained liquid fuel combustion.

Fig 25 shows the NOx variation curves under different injection angles. Post-DME ignition at $t = 0.36$ ms, NOx concentrations ascend with angle-dependent gradients: steepest slopes at $\beta = 60°$ during $t = 0.36$–1.1 ms, progressively flattening toward $\beta = 180°$. Crucially, sustained NOx formation persists beyond $t = 1.1$ ms at $\beta = 180°/150°$, whereas profiles plateau at acute angles ($\beta \leq 120°$). Cross-referencing OH/NH$_2$ profiles in Figs 19 and 20 confirms superior ammonia ignition efficacy

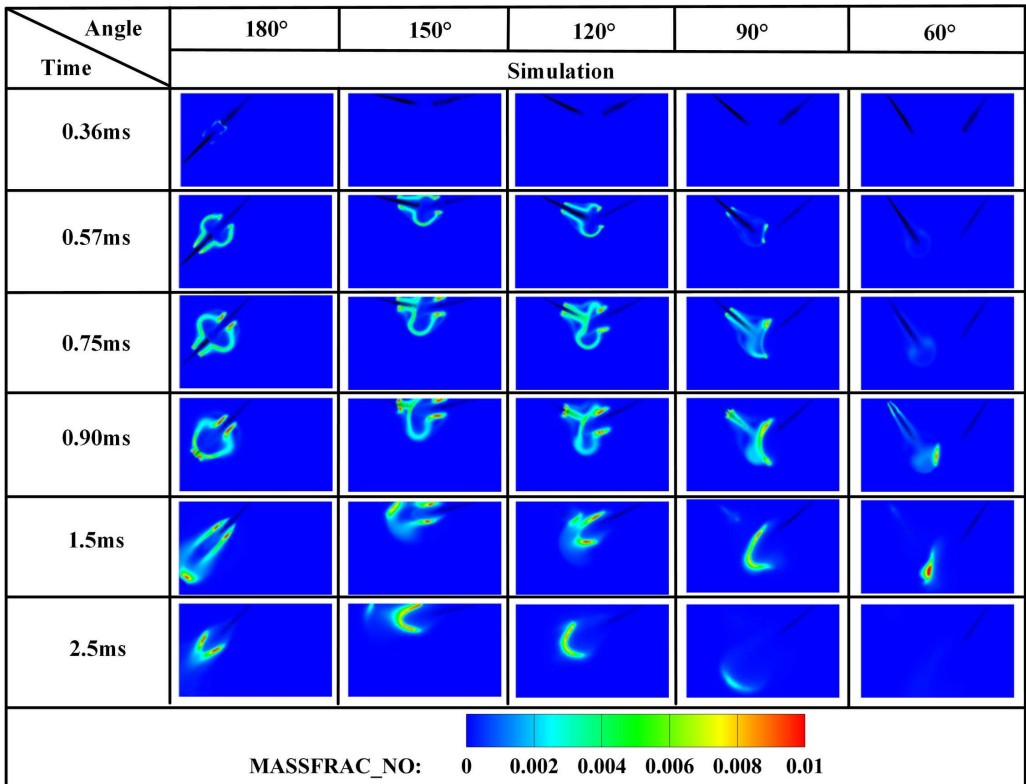

**Fig 21. Spatial distribution of NO radicals.**

at $\beta = 180°$. This demonstrates that wider angles shorten fuel contact duration and compromise atomization quality, yet enable optimal thermal coupling during peak DME combustion. Conversely, acute angles prolong pre-ignition mixing, delaying high-temperature exposure and impairing combustion completeness.

Figs 26 and 27 presents an analysis of the effects of different injection angles on ignition combustion and pollutant formation. Fig 26 illustrates the ignition delay period, combustion duration, and end of combustion under various injection angles. It can be observed that the ignition delay period is shortest at an injection angle of 90°, while the longest delay occurs at 150°. This is primarily because a lower injection angle ($\beta = 60°$) results in delayed contact between the DME flame and the liquid ammonia, where the flame temperature is insufficient to ignite the ammonia early. In contrast, a larger injection angle ($\beta = 150°$) leads to incomplete atomization of liquid ammonia. When the poorly atomized ammonia interacts with the DME flame, it undergoes a heat-absorbing vaporization process, thereby prolonging the ignition delay. At $\beta = 180°$, the contact time between DME and liquid ammonia is shortest. The DME flame envelops the ammonia spray and propagates forward along with its trajectory, resulting in a slightly shorter ignition delay compared to $\beta = 150°$. The longest combustion duration occurs at an injection angle of $\beta = 120°$, followed by $\beta = 150°$ and 180°. This can be attributed to the fact that at $\beta = 120°$, the atomization of liquid ammonia is more effective than at 150° and 180°, while the interaction time between DME and ammonia is longer. These conditions promote better early-stage combustion, leading to an extended main combustion phase. However, toward the end of combustion, a declining trend in heat release rate is observed. As can be seen from the NOx production shown in Fig 27, the highest NOx emissions occur at an injection angle of β = 60°, while the lowest are observed at β = 120°. This trend corresponds well with the combustion behavior illustrated in Fig 26.

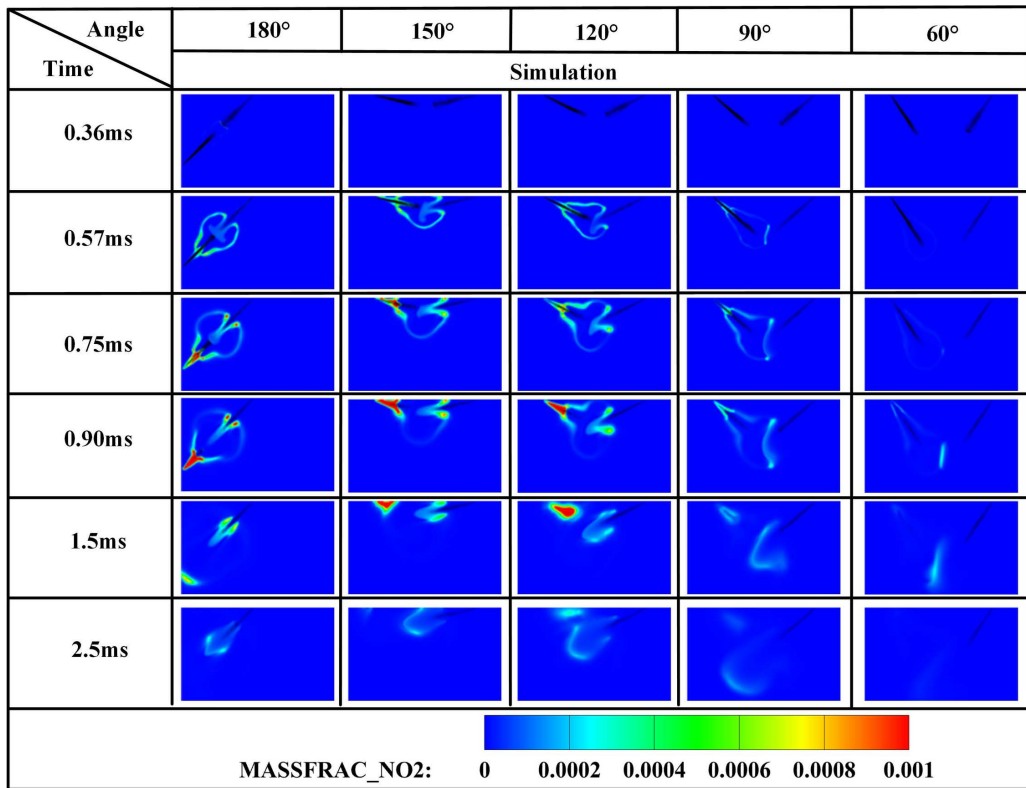

**Fig 22. Spatial distribution of NO₂ radicals.**

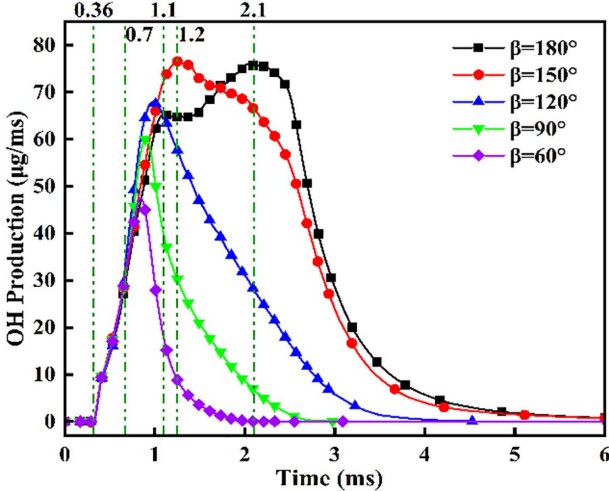

**Fig 23. OH concentration profiles across varying fuel injector angles.**

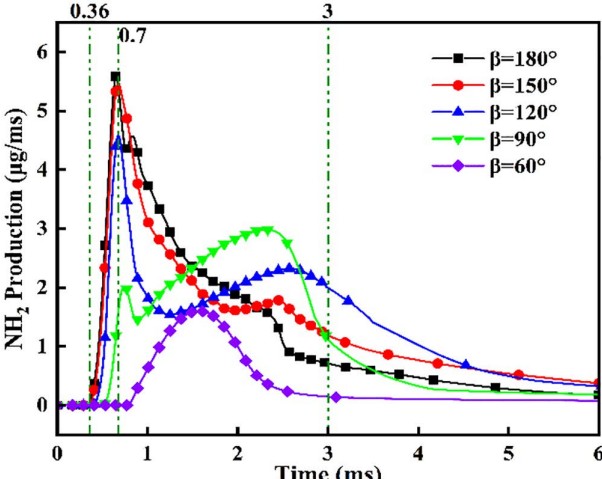

**Fig 24. NH₂ radical concentration profiles across varying fuel injector angles.**

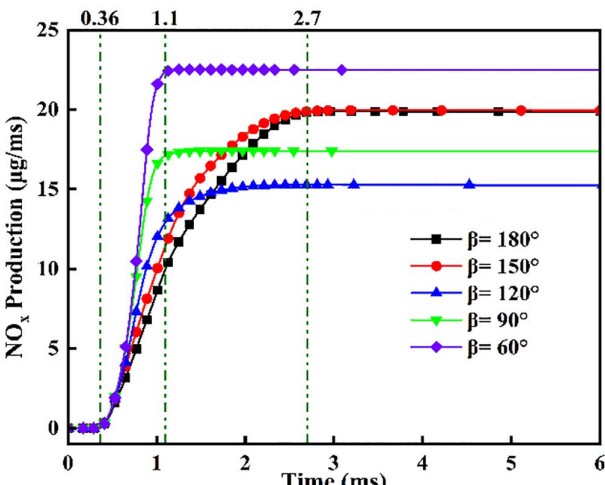

**Fig 25. Temporal evolution of NOx concentration across varying fuel injector angles.**

Figs 28 and 29 depict maximum temperatures and heat release rates under fixed ammonia energy share ($\alpha = 80\%$) and injector spacing ($L = 6\,cm$) across varied injection angles. Fig 28 reveals rapid temperature ascent to ~2700 $K$ post-DME ignition at $t = 0.36\,ms$, sustained through $t = 0.7\,ms$ during continuous DME injection. This thermal plateau correlates with DME's 350°C ignition point and 0.7 ms injection duration. Subsequent bifurcated cooling stems from evaporative heat absorption during flame-ammonia interaction post-injection. Crucially, during $t = 0.7–1.0\,ms$, $\beta = 60°$ maintains peak temperatures due to enhanced atomization promoting vapor-phase combustion, yet exhibits the most rapid subsequent decline from reignition of recirculated fuel missing peak thermal conditions. Conversely, $\beta = 180°/150°$ demonstrates moderated cooling rates attributable to optimal thermal coupling during DME's peak combustion intensity. Fig 29 confirms progressively reduced heat release rates with decreasing angles: minimum at $\beta = 60°$, marginally higher at $\beta = 150°$ versus $\beta = 180°$, yet collectively exceeding narrower angles ($\beta \leq 120°$) by 18–25% in cumulative output.

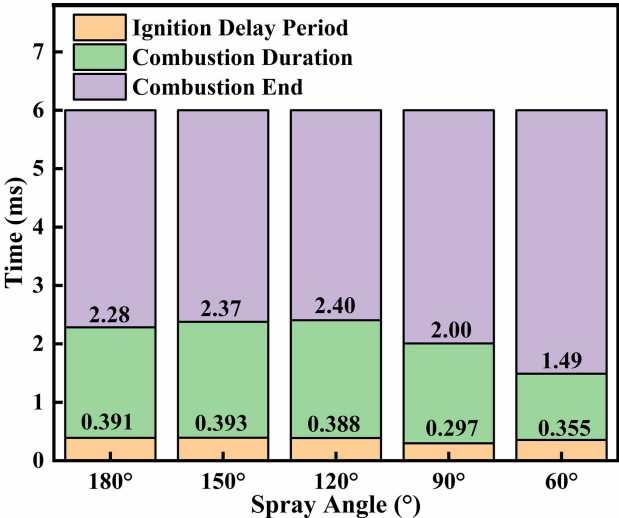

**Fig 26. Combustion phasing under different injection angles.**

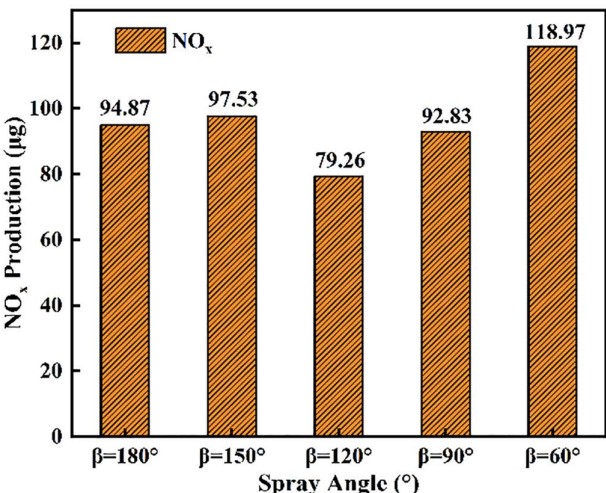

**Fig 27. Cumulative NOx production across varying fuel injector angles.**

## 4.3. Analysis of ignition-combustion and pollutant formation characteristics with varying ammonia energy shares in liquid ammonia-DME dual-fuel mode

To maximize ammonia energy share while minimizing pollutant emissions, this study investigates liquid ammonia energy fractions ($\alpha_{NH_3}$) of 70%, 80%, and 90% under optimized geometric parameters: inter-injector distance $L = 6$ cm and injection angle $\beta = 180°$, as established in prior analyses.

Fig 30 displays temperature distributions under varying ammonia energy shares. At $\alpha = 70\%$, DME ignition occurs at $t = 0.36$ ms with flames propagating radially, enveloping the liquid ammonia spray. Conversely, at $\alpha = 80\%$, initial fuel contact develops at t = 0.36 ms without complete flame encapsulation. Notably, at α = 90%, DME ignition delays until $t = 0.57$ ms, followed by gradual flame propagation surrounding ammonia spray. This phenomenon arises from ammonia's

**Fig 28. Maximum temperature.**

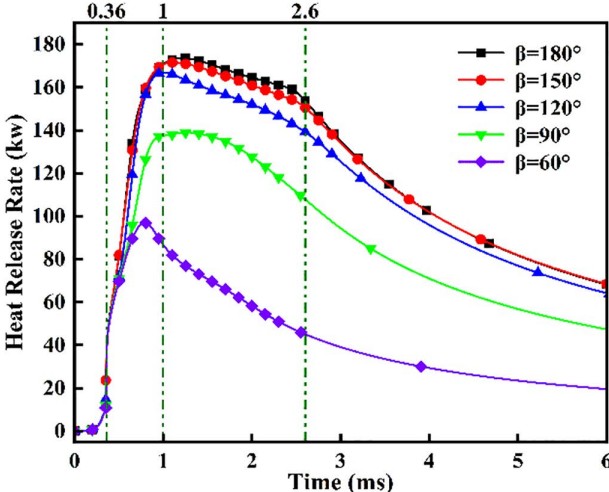

**Fig 29. Heat Release Rate.**

inherent properties—high vaporization enthalpy (1370 *kJ/kg*) and elevated autoignition temperature (>900 *K*)—combined with energy allocation effects quantified in Table 6: A higher DME energy fraction at $\alpha = 70\%$ enables more vigorous combustion, whereas marginal DME content at $\alpha = 90\%$ compromises ignition efficacy despite meeting minimum ignition requirements.

Figs 31–34 present OH, NH$_2$, NO, and NO$_2$ distributions. Fig 31 reveals OH radical generation commencing at $t = 0.36$ ms for $\alpha = 70\%$, where ammonia jet impingement induces distinct flame segmentation. At $\alpha = 80\%$, attenuated impingement occurs at identical timing without complete segmentation. For $\alpha = 90\%$, OH emergence delays until $t = 0.57$ ms with premature quenching by $t = 2.5$ ms, indicating combustion cessation. This progressive reduction in OH distribution area with increasing ammonia share stems from diminished DME energy fraction: insufficient fuel mass fails to achieve reliable ignition at the 950 *K* ambient condition at $t = 0.36$ ms, compromising pilot efficacy. Fig 32 corroborates

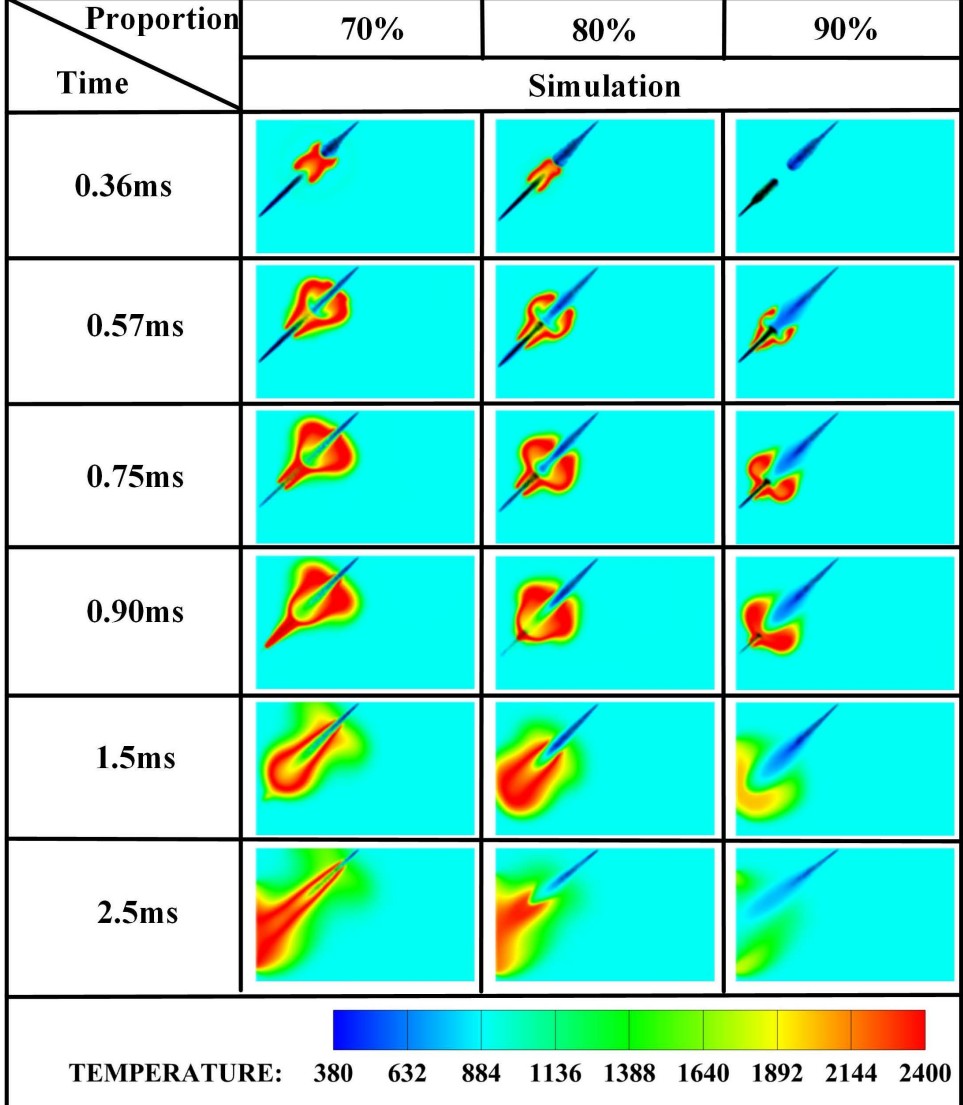

**Fig 30. Temperature distributions under varying liquid ammonia energy shares.**

delayed $NH_2$ emergence ($t=0.36$ ms for $α=70\%$, attenuated intensity at $α=80\%$, $t=0.51$ ms for $α=90\%$) and earlier extinction at higher ammonia shares, confirming deteriorating combustion efficiency. Figs 33 and 34 demonstrate consistent $NO/NO_2$ reduction with increasing α, aligning with literature [38]. This dual mechanism involves (1) an enriched DME fraction elevating H,O,OH radicals that promote thermal-NO and (2) enhanced $NH_2$ availability for NO reduction via Reaction (9):

$$NO + NH_2 \longleftrightarrow N_2 + H_2O \qquad (9)$$

Figs 35−38 present the combustion characteristics curves of OH, $NH_2$, and NOx, respectively, under different ammonia energy fractions. Fig 35 reveals consistently diminished OH production at $α=90\%$, peaking at $t=1.0$ ms with 25–30% lower magnitude versus $α=70\%/80\%$. Corroborating Fig 17a contours showing absent OH by $t=2.5$ ms at $α=90\%$, this

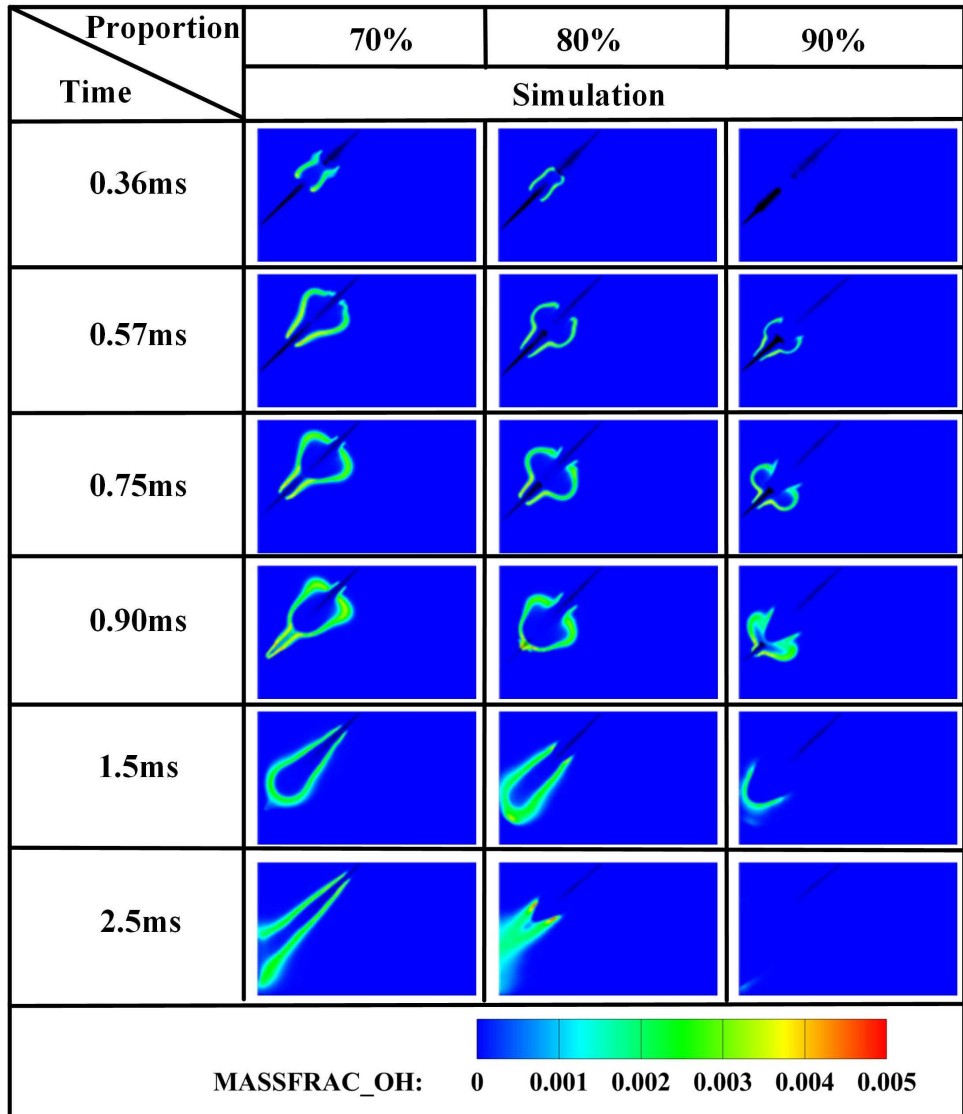

**Fig 31. Spatial distribution of OH radicals.**

confirms compromised pilot ignition efficacy at high ammonia fractions. For $\alpha = 80\%$, a plateau phase emerges during $t = 1.1$–$1.5$ ms where sustained OH generation reflects DME-initiated ammonia ignition, followed by accelerated production during rapid combustion until fuel depletion reduces rates. The $\alpha = 70\%$ profile exhibits a biphasic response: an initial peak at $t = 1.0$ ms signifying DME-dominated combustion, a subsequent decline from evaporative cooling during ammonia entrainment, and a distinct secondary rise peaking at $t = 1.5$ ms indicating delayed autoignition of ammonia post-DME combustion.

Fig 36 demonstrates that at $\alpha = 70\%$, the highest DME energy fraction drives the earliest $NH_2$ concentration rise, culminating in peak intensity near $t = 0.7$ ms. The superior $NH_2$ peak magnitude at $\alpha = 80\%$ reflects optimal ammonia ignition efficacy during DME's combustion phase. Conversely, $\alpha = 90\%$ exhibits delayed $NH_2$ ascent and 30–35% lower peak values, indicating compromised ignition due to excessive ammonia fraction. All configurations achieve maximum $NH_2$

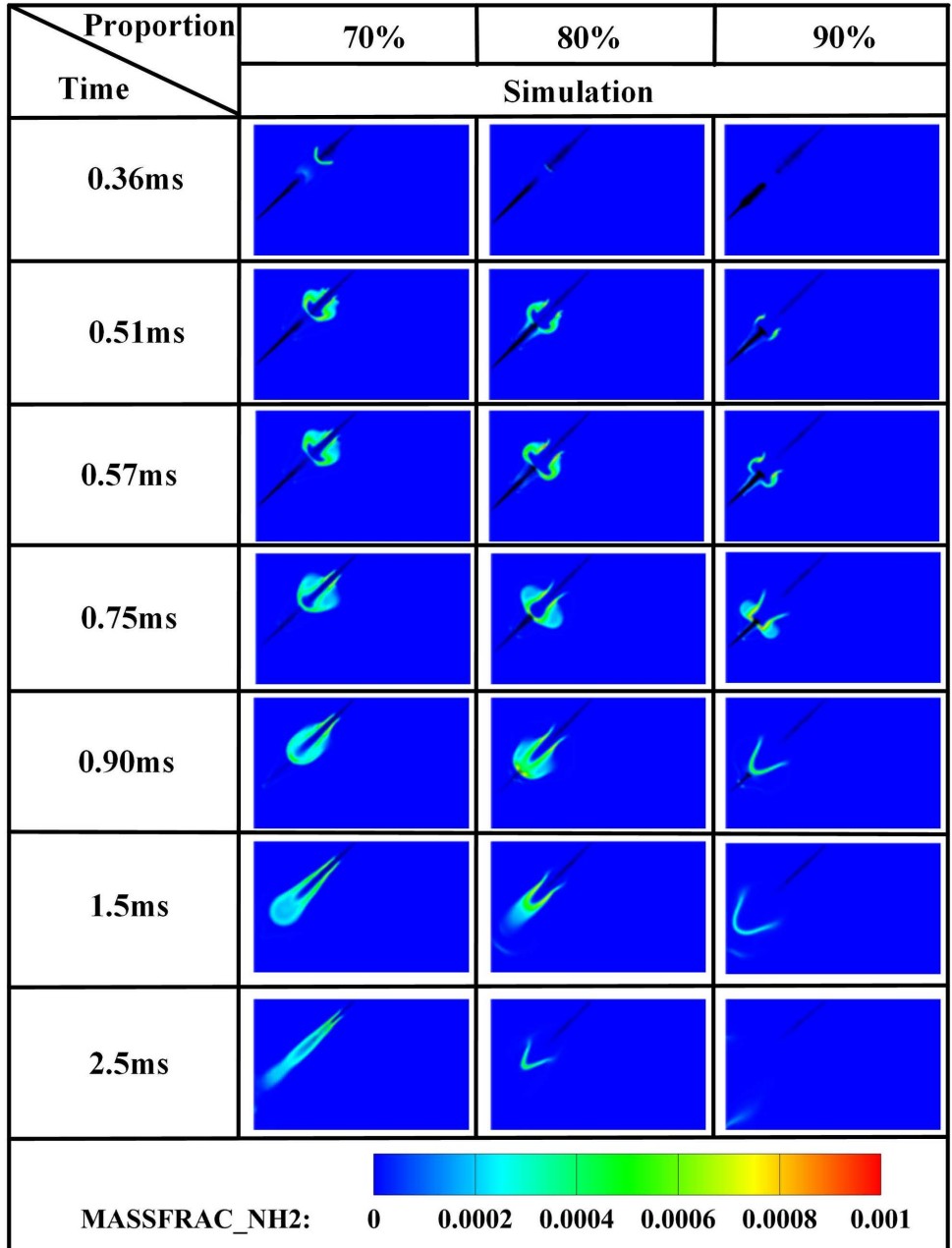

**Fig 32. Spatial distribution of NH₂ radicals.**

concentrations near $t = 0.7\,\text{ms}$, synchronized with DME injection termination and fuel interaction. The bimodal profiles at $\alpha = 80\%/90\%$ reveal dual-phase dynamics: the primary peak ($t = 0.7\,\text{ms}$) signifies flame-entrained partial oxidation, followed by secondary peaks indicating delayed autoignition with diminished reaction rates. While $\alpha = 70\%$ and $\alpha = 90\%$ show 18–22% lower peak $NH_2$ production versus $\alpha = 80\%$, post-$t = 1.2\,\text{ms}$ concentrations at $\alpha = 70\%$ exceed those at $\alpha = 80\%$. This transition arises from autoignition dominance after DME depletion, evidenced by the $\alpha = 70\%$ profile's resurgence phase peaking post-$t = 1.5\,\text{ms}$.

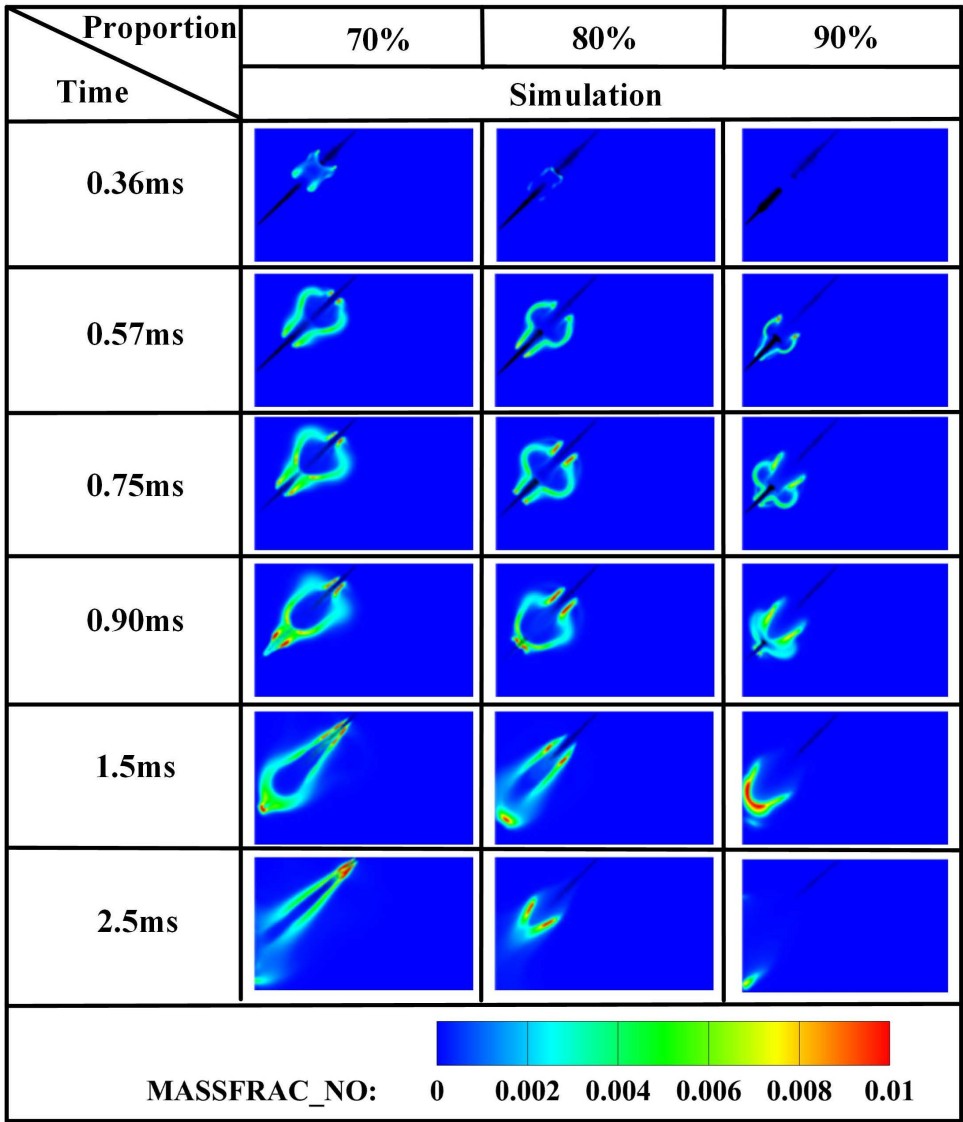

Fig 33. Spatial distribution of NO radicals.

Fig 37 reveal NOx concentration ascent commencing at $t = 0.36$ ms post-DME ignition with distinct formation gradients: steepest at $\alpha = 70\%$, intermediate at $\alpha = 80\%$, and gentlest at $\alpha = 90\%$. Plateau timing exhibits strong ammonia share dependence: $t = 3.0$ ms ($\alpha = 70\%$), $t = 2.7$ ms ($\alpha = 80\%$), and $t = 1.1$ ms ($\alpha = 90\%$). This graph demonstrates increased NOx yield at lower ammonia fractions, attributable to higher DME energy content driving more vigorous combustion. Conversely, excessive ammonia ($\alpha = 90\%$) provides insufficient DME for reliable ignition. Consequently, ammonia energy share critically governs combustion outcomes: moderate fractions ($\alpha = 80\%$) optimize OH/NH$_2$ generation, while low fractions ($\alpha = 70\%$) elevate NOx production by 25–40% versus $\alpha = 80\%$.

Figs 38 and 39 presents an analysis of the effects of different ammonia energy fractions on ignition combustion and pollutant formation. Fig 38 illustrates the ignition delay period, combustion duration, and end of combustion under various ammonia energy fractions. It can be observed that the ignition delay period is shortest at an ammonia energy

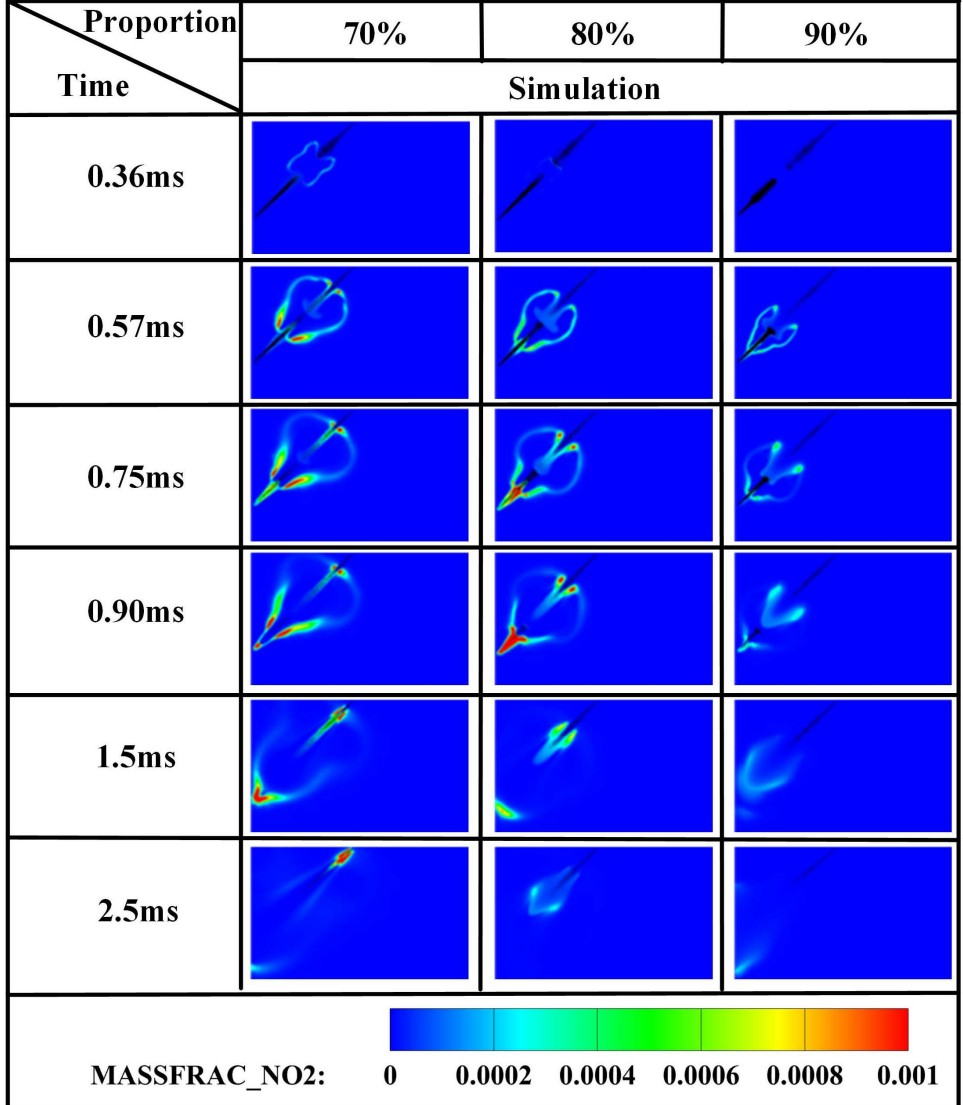

| Proportion / Time | 70% | 80% | 90% |
|---|---|---|---|
| | Simulation | | |
| 0.36ms | | | |
| 0.57ms | | | |
| 0.75ms | | | |
| 0.90ms | | | |
| 1.5ms | | | |
| 2.5ms | | | |
| MASSFRAC_NO2: | 0    0.0002   0.0004   0.0006   0.0008   0.001 | | |

**Fig 34. Spatial distribution of NO$_2$ radicals.**

fraction of $\alpha = 70\%$, and longest at $\alpha = 90\%$. This behavior can be attributed to the fact that at $\alpha = 70\%$, the corresponding DME energy fraction is higher, leading to more intense combustion of the DME flame and thus a shorter ignition delay. In contrast, at $\alpha = 90\%$, the reduced DME energy fraction results in weaker flame reactivity, prolonging the ignition delay. Regarding the combustion duration, the longest period occurs at $\alpha = 70\%$, followed by $\alpha = 80\%$, while the shortest duration is observed at $\alpha = 90\%$. This trend shows an inverse correlation with the ignition delay: a lower ammonia energy fraction corresponds to a higher DME energy fraction, promoting more vigorous flame combustion and thereby extending the combustion duration. Similarly, Fig 39 indicates that a decrease in the ammonia energy fraction leads to an increase in the DME energy fraction, resulting in more intense combustion and consequently higher NOx emissions. Conversely, a higher ammonia energy fraction reduces the DME proportion, which is insufficient to ensure effective ignition of the ammonia.

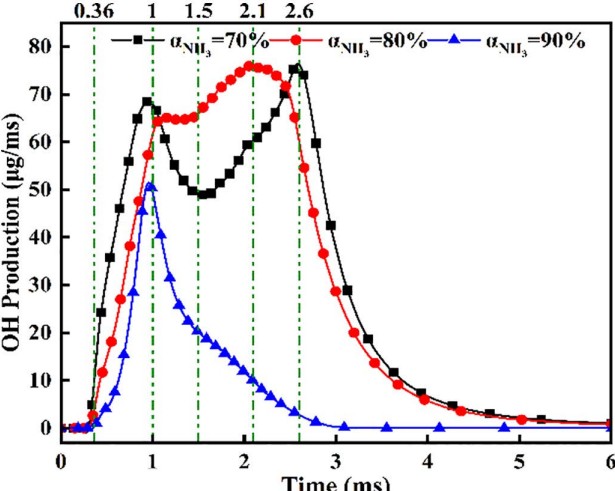

**Fig 35. Temporal profiles of OH radicals across ammonia energy fractions.**

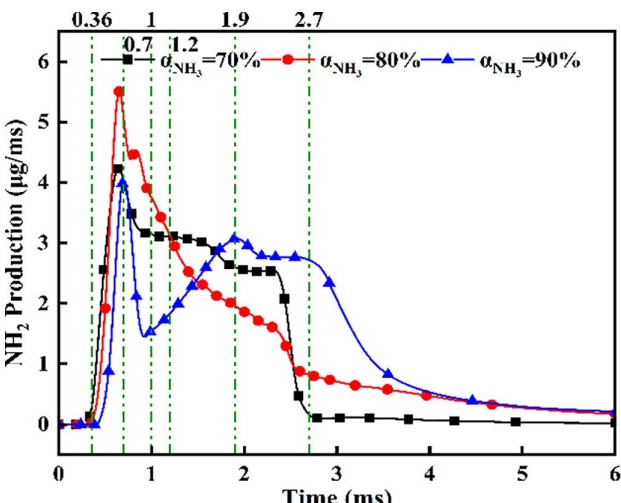

**Fig 36. Temporal profiles of NH$_2$ radicals across ammonia energy fractions.**

Figs 40 and 41 depict maximum temperatures and heat release rates under simultaneous dual-injection ($\beta = 180°$, $L = 6$ cm) for varied ammonia energy shares. Fig 40 shows a rapid temperature rise to ~2700 $K$ post-DME ignition at $t = 0.36$ ms, with bifurcated cooling commencing post-$t = 0.7$ ms due to evaporative heat absorption during flame-ammonia interaction. Crucially, during $t = 0.7$–1.1 ms, $\alpha = 90\%$ maintains 80–120 K higher temperatures than other shares, yet undergoes precipitous decline post-$t = 1.1$ ms from insufficient DME energy for sustained combustion. Conversely, $\alpha = 70\%$ exhibits superior temperature retention post-$t = 1.1$ ms, with accelerated cooling after $t = 2.7$ ms indicating near-complete fuel depletion. Corroborating Fig 41, $\alpha = 70\%$ achieves the highest increase in heat release rate post-$t = 0.36$ ms. All profiles exhibit characteristic dual-phase ignition: initial heat release decline stems from ammonia's vaporization enthalpy (1370 $kJ/kg$), with decreasing magnitudes and extended combustion durations at higher ammonia shares. Specifically, total heat release reduces by 18−25%, while combustion persistence increases 0.3–0.5 ms per 10% ammonia increment.

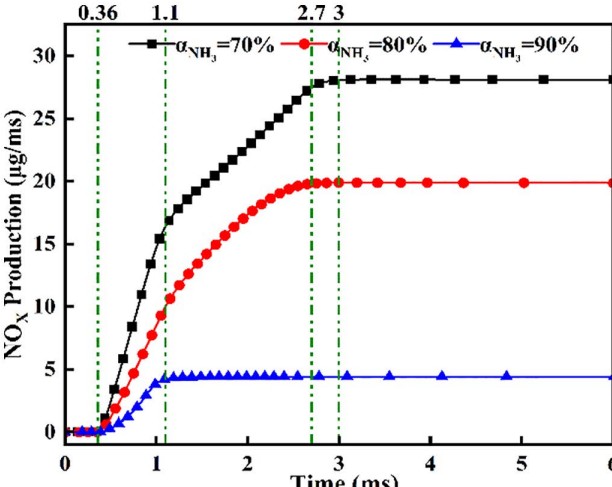

**Fig 37. Temporal evolution of NOx concentration across ammonia energy fractions.**

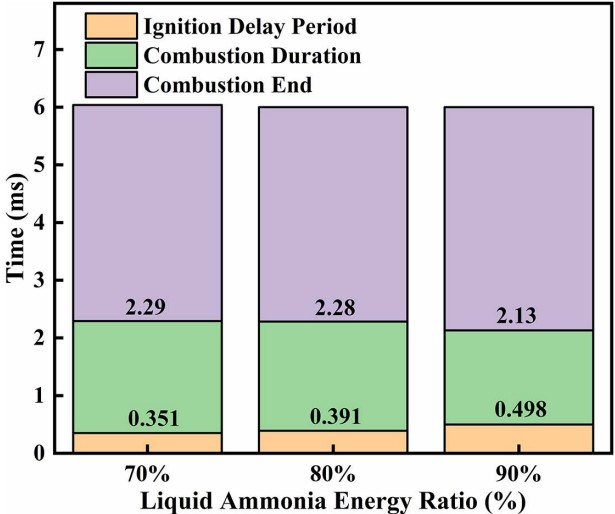

**Fig 38. Combustion phasing under different ammonia energy fractions.**

## 5. Conclusions

This study demonstrated that a flexible ammonia-DME dual-fuel high-pressure direct injection strategy can effectively overcome ammonia's high ignition energy and slow combustion rate. The main conclusions are as follows:

(1) The configuration with injector spacing $L = 6$ cm and injection angle 180° achieves optimal dimethyl ether (DME)-initiated ammonia ignition, exhibiting peak heat release rate. Conversely, larger spacings ($L = 7/8$ cm) or reduced injection angles (150°/120°/90°/60°) degrade ignition performance. This deterioration occurs because although increased spacing/angle enhances fuel-air and inter-fuel mixing, it simultaneously extends the DME flame propagation timescale to ammonia contact. During this delay, accelerated temperature decay in the DME combustion zone due to heat absorption by liquid ammonia spray prevents successful ammonia ignition. Concurrently, reduced injector spacing

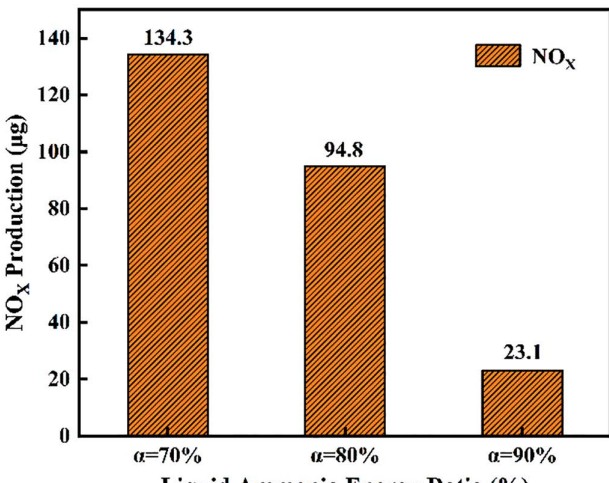

**Fig 39. Cumulative NOx production across ammonia energy fractions.**

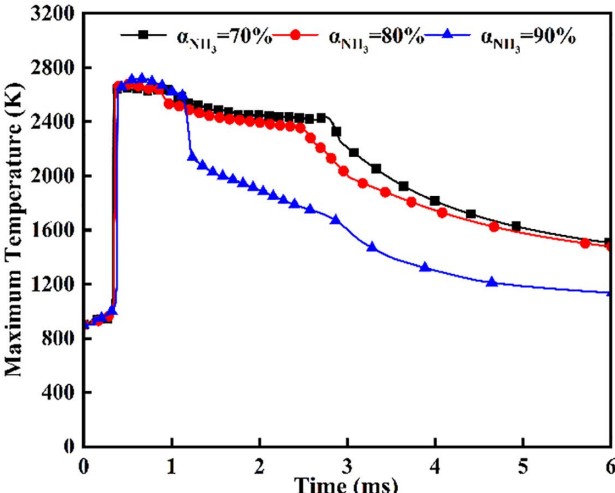

**Fig 40. Maximum temperature.**

under DME combustion conditions diminishes OH distribution areas while enhancing $NH_2$ fields. This signifies intensified thermal suppression effects from ammonia entrainment and high vaporization enthalpy (1370 *kJ/kg*), coupled with greater pilot-fuel involvement in ammonia ignition. Furthermore, during DME-inactive phases, both OH and $NH_2$ spatial extents increase at shorter spacings, demonstrating earlier fuel interaction, expanded reaction zones, and improved combustion completeness through optimized air-fuel mixing.

(2) Optimal dimethyl ether (DME)-initiated ammonia ignition is achieved at 6 cm injector spacing. For configurations with 7–8 cm spacings, the $NH_2$ concentration profile with two peaks—characterized by an initial decline succeeded by a secondary rise—stems from competing mechanisms. This phenomenon arises from two competing effects of increased injector spacing: The prolonged flame-to-ammonia propagation time results in temperature inadequacy

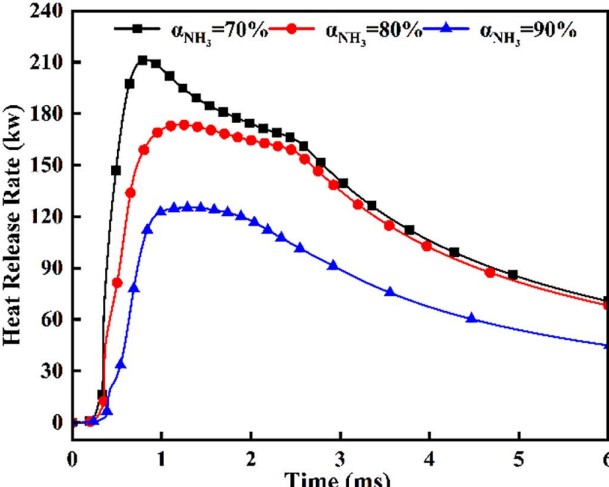

**Fig 41. Heat Release Rate.**

in the DME combustion zones, while the improved atomization quality expands the prevaporized ammonia regions, thereby facilitating flame-front ignition. Given constant total ammonia mass flow, the bimodal $NH_2$ profiles at L=7–8 cm spacings reveal distinct combustion phases: the first peak signifies combustion of flame-entrained liquid ammonia droplets within the DME flame front, peaking at 0.7 ms before declining, while the secondary peak indicates autogenous ammonia ignition characterized by attenuated reaction kinetics. This experiment demonstrates that increased injector spacing prolongs DME-ammonia interaction time, progressively reduces heat release rate by 19.2±0.7% per cm increment, and deteriorates ignition efficacy.

(3) Analysis of injection angle effects on DME-initiated ammonia combustion reveals superior ignition performance at 180° and 150° configurations compared to narrower angles (120°/90°/60°). Between the two optimal angles, the 150° case demonstrates higher peak concentrations in both OH and $NH_2$ temporal profiles. However, heat release rate (HRR) analysis shows marginally greater magnitude at 180° with narrow disparity (<5%) between these configurations.

(4) In DME-initiated ammonia combustion, NO formation predominantly concentrates in high-temperature zones (>1800 K), while $NO_2$ generation primarily occurs in medium-to-low temperature regions (800–1500 K) with substantially lower yield. Paradoxically, reducing the ammonia energy fraction from 90% to 70% increases both NO and $NO_2$ production. Concurrently, this reduction enhances DME-assisted ignition effectiveness, evidenced by elevated peak heat release rates in HRR profiles. The 70% ammonia energy fraction configuration achieves maximum HRR magnitude, indicating optimal combustion intensity.

## Author contributions

**Conceptualization:** Jun Fu, Han He.

**Data curation:** Han He.

**Formal analysis:** Jun Fu.

**Funding acquisition:** Han He, Feibin Yan, Yi Ma.

**Investigation:** Han He, Feibin Yan, Shuhui Wang.

**Methodology:** Feibin Yan.

**Project administration:** Jun Fu.

**Resources:** Han He, Feibin Yan, Yi Ma.

**Software:** Shuhui Wang.

**Supervision:** Feibin Yan.

**Validation:** Jun Fu, Yi Ma.

**Visualization:** Yi Ma.

**Writing – original draft:** Han He.

**Writing – review & editing:** Jun Fu, Shuhui Wang.

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
