## [Decision Letter · Decision Letter 0]

11 Aug 2025

Dear Dr. Fu,

We look forward to receiving your revised manuscript.

Kind regards,

Krishnamoorthy Ramalingam

Academic Editor

PLOS ONE

Journal Requirements:

“This research was supported by the China Postdoctoral Science Foundation (Grant No. 2024M760170), the Hunan Provincial Natural Science Foundation of China (Grant No. 2023JJ50262), and the Hunan Provincial Department-level Project (Grant No. LXBZZ2024391).”

“This research was supported by the China Postdoctoral Science Foundation (Grant No. 2024M760170), the Hunan Provincial Natural Science Foundation of China (Grant No. 2023JJ50262), and the Hunan Provincial Department-level Project (Grant No. LXBZZ2024391).”

“This research was supported by the China Postdoctoral Science Foundation (Grant No. 2024M760170), the Hunan Provincial Natural Science Foundation of China (Grant No. 2023JJ50262), and the Hunan Provincial Department-level Project (Grant No. LXBZZ2024391).”

5. We note that your Data Availability Statement is currently as follows: All relevant data are within the manuscript and its Supporting Information files.

Reviewers' comments:

Reviewer's Responses to Questions

**Comments to the Author**

1. Is the manuscript technically sound, and do the data support the conclusions?

Reviewer #1: Yes

Reviewer #2: Yes

2. Has the statistical analysis been performed appropriately and rigorously?

Reviewer #1: Yes

Reviewer #2: Yes

3. Have the authors made all data underlying the findings in their manuscript fully available?

Reviewer #1: Yes

Reviewer #2: Yes

4. Is the manuscript presented in an intelligible fashion and written in standard English?

Reviewer #1: Yes

Reviewer #2: No

Reviewer #1: This manuscript investigates the spray combustion characteristics of liquid ammonia/dimethyl ether (DME) under high-pressure dual direct injection strategies, demonstrating significant academic value and engineering application potential. Through numerical simulations, the study systematically examines the effects of injector spacing, injection angle, and ammonia energy ratio on combustion characteristics, yielding several meaningful conclusions. The paper is well-structured and methodologically sound, but the following aspects require further improvement:

1. While the study focuses on numerical simulations of fundamental spray combustion characteristics under engine conditions, the existing literature review leans more toward engine research. Expanding the discussion on advancements in fundamental spray combustion studies would be beneficial.

2. The parameter settings for key sub-models are incompletely described, particularly the critical parameters of the droplet breakup model (e.g., KH-RT model) for high-pressure liquid ammonia injection. The high latent heat of vaporization of ammonia and the low boiling point of DME make the breakup and evaporation processes crucial in determining combustion characteristics. Improper parameter settings may lead to deviations in key metrics such as spray penetration length and SMD, thereby affecting the reliability of conclusions regarding ignition delay and flame stability.

3. The caption of Figure 7 is incomplete, and 'Figure 7(a)' is missing. Please check the figure and table information in the manuscript.

Reviewer #2: The manuscript investigates spray combustion characteristics of liquid ammonia and dimethyl ether (DME) dual-fuel systems under various injection strategies using CFD simulations. The authors explore the effects of injector spacing (6–8 cm), injection angle (60°–180°), and ammonia energy share (70–90%) on ignition, combustion, radical formation (OH, NH₂), and NOx emissions. The KH–RT breakup model and SAGE detailed chemistry model are employed to simulate spray and combustion behavior, with model validation performed against constant-volume combustion vessel (CVCV) experimental data for both ammonia and DME sprays. The main findings suggest that 6 cm injector spacing, 180° injection angle, and 80% ammonia energy share provide optimal ignition and combustion performance, while balancing NOx emissions. Shorter spacing enhances early fuel interaction and ammonia combustion completeness; wider angles promote earlier ignition; and moderate ammonia fractions achieve better OH/NH₂ generation without excessive NOx formation.

1. The spray models for ammonia and DME are validated individually, but no validation is provided for the combined dual-fuel injection combustion cases. Without this, the predictive accuracy for interaction phenomena remains uncertain.

2. Grid independence is shown for ammonia spray only. The authors should also demonstrate that mesh resolution is sufficient for dual-fuel reacting cases, where flame fronts and radical fields require higher fidelity.

3. The results section relies heavily on descriptive comparisons of figures. More quantitative metrics (ignition delay, flame length, total heat release, integrated NOx mass, etc.) should be systematically presented in tables and statistically analyzed.

4. It is not specified whether the simulations use EDC, presumed PDF, or another turbulence–chemistry coupling model for the SAGE solver. This omission limits reproducibility.

5. Critical parameters like wall temperature, pressure history, and ambient composition control during combustion are not fully detailed.

6. The model geometry for dual injectors (nozzle hole size, orientation details, distance to ignition region) should be clearly illustrated with scaled schematics.

7. The results would be stronger if uncertainties from model parameters (spray breakup constants, kinetic rates) were quantified through sensitivity analysis.

8. The introduction contains extended summaries of many unrelated ammonia pilot-fuel studies, which could be condensed to highlight the specific knowledge gap this work addresses (i.e., injector spacing and angle effects for NH₃/DME).

**Do you want your identity to be public for this peer review?** For information about this choice, including consent withdrawal, please see our Privacy Policy

Reviewer #1: No

Reviewer #2: No

---

## [Author Response · Author response to Decision Letter 1]

29 Aug 2025

Dear Editors:

Thanks for your letter concerning our manuscript entitled ' Study on Spray Combustion Characteristics of Liquid Ammonia/Dimethyl Ether Dual Fuel Based on Different Injection Strategies ' (Manuscript Number: PONE-D-25-39351). Those comments are all valuable and very helpful for revising and improving our manuscript and significantly guide our research. We have studied these comments carefully and made modifications which we hope meet with your approval. Revised portions are marked red in the manuscript. The main corrections in the manuscript and the responses to the reviewers' comments are listed below.

[Comment 1]: While the study focuses on numerical simulations of fundamental spray combustion characteristics under engine conditions, the existing literature review leans more toward engine research. Expanding the discussion on advancements in fundamental spray combustion studies would be beneficial.

Response: Thank you for the reviewer 's comments. The introduction of this paper increases the basic research progress of spray combustion of ammonia and dimethyl ether and their dual fuels. The literature review of spray combustion of liquid ammonia fuel shows that it requires high thermal atmosphere conditions ( ambient temperature 850 K and ambient density 21 % ) to ignite and burn, and the combustion speed is slow, indicating that liquid ammonia fuel has difficulties in ignition and combustion instability. The literature review of dimethyl ether fuel spray combustion shows that it has good ignition and combustion characteristics, and its ignition delay period is short. It is an excellent high-activity ignition fuel.

[Comment 2]: The parameter settings for key sub-models are incompletely described, particularly the critical parameters of the droplet breakup model (e.g., KH-RT model) for high-pressure liquid ammonia injection. The high latent heat of vaporization of ammonia and the low boiling point of DME make the breakup and evaporation processes crucial in determining combustion characteristics. Improper parameter settings may lead to deviations in key metrics such as spray penetration length and SMD, thereby affecting the reliability of conclusions regarding ignition delay and flame stability.

Response: Thanks for the reviewer’s comments. The parameter setting of the key sub-model is not fully described. In this paper, six breaking constants in the KH-RT breaking model are re-added to ensure the reliability of the conclusion.

[Comment 3]: The caption of Figure 7 is incomplete, and 'Figure 7(a)' is missing. Please check the figure and table information in the manuscript.

Response: Thanks for the reviewer’s comments. Missing information was checked and added in the revised version.

[Comment 4]: The spray models for ammonia and DME are validated individually, but no validation is provided for the combined dual-fuel injection combustion cases. Without this, the predictive accuracy for interaction phenomena remains uncertain.

Response: Thank you for the reviewer 's comments. At present, there is no experimental data on DME and liquid ammonia dual-fuel compression ignition. However, the model establishment part of this paper refers to the ' Numerical Study on High-Pressure Direct Injection Combustion Characteristics of Ammonia / Dimethyl Ether Dual Fuel ' of Chinese Internal Combustion Engine Engineering. Article number : 1000-0925 ( 2024 ) 04-0016-13. The reference verifies the spray models of ammonia and dimethyl ether respectively, thus completing the verification of the numerical model.

[Comment 5]: Grid independence is shown for ammonia spray only. The authors should also demonstrate that mesh resolution is sufficient for dual-fuel reacting cases, where flame fronts and radical fields require higher fidelity.

Response: Thanks for the reviewer’s comments. Fig 2 compares the simulation results with the simulation and experimental spray patterns of Li et al., where the black solid line represents the experimental liquid phase ammonia spray and the black dotted line represents the experimental gas phase ammonia spray. The results show that the spray model can accurately predict the spray morphology of liquid ammonia, and the applicability of the model in numerical simulation is verified.

[Comment 6]: The results section relies heavily on descriptive comparisons of figures. More quantitative metrics (ignition delay, flame length, total heat release, integrated NOx mass, etc.) should be systematically presented in tables and statistically analyzed.

Response: Thanks for the reviewer’s comments. Under the conditions of different injection distances, different injection angles and different energy ratios of liquid ammonia, the ignition delay period, combustion duration and combustion end period of DME igniting liquid ammonia are presented in the form of tables, and the production of NOx under different conditions is used to better analyze the effect of DME igniting liquid ammonia under different conditions. Its related content has been updated in the revised version of the article.

[Comment 7]: It is not specified whether the simulations use EDC, presumed PDF, or another turbulence–chemistry coupling model for the SAGE solver. This omission limits reproducibility.

Response: Thanks for the reviewer’s comments. For the SAGE solver, I use the SAGE presumed PDF model to calculate the chemical reaction heat release and molecular diffusion process during fuel combustion. The model is based on the theory of chemical reaction kinetics, which can calculate the reaction rate of each basic reaction in the reaction mechanism and solve the transport equation. At the same time, the RAG k-ε model in the RANS model is selected, which has high calculation accuracy and efficiency, and can ensure that the calculation results have more application value. Its content has also been supplemented in the revised version.

[Comment 8]: Critical parameters like wall temperature, pressure history, and ambient composition control during combustion are not fully detailed.

Response: Thanks for the reviewer’s comments. The key parameters ( wall temperature, ambient pressure, ambient density and oxygen content ) in the combustion process are shown in Table 5.

[Comment 9]: The model geometry for dual injectors (nozzle hole size, orientation details, distance to ignition region) should be clearly illustrated with scaled schematics.

Response: Thanks for the reviewer’s comments. The schematic diagram of the double injector has been updated, as shown in Figure 5, and the injector and related position dimensions are explained in Table 4. In addition, dimethyl ether ignites liquid ammonia by means of compression ignition, so the injector has no distance to the ignition area.

[Comment 10]: The results would be stronger if uncertainties from model parameters (spray breakup constants, kinetic rates) were quantified through sensitivity analysis.

Response: Thanks for the reviewer’s comments. I have quantified the model parameters to ensure the reliability of the conclusions. At the same time, the quantitative content of model parameters has been reflected in the revised version.

[Comment 11]:The introduction contains extended summaries of many unrelated ammonia pilot-fuel studies, which could be condensed to highlight the specific knowledge gap this work addresses (i.e., injector spacing and angle effects for NH₃/DME).

Response: Thanks for the reviewer’s comments. The research on ammonia fuel independence has been streamlined and the research on dual-fuel combustion of dimethyl ether and ammonia has been updated. However, there are few studies on the dual-fuel high-pressure direct injection combustion of dimethyl ether and liquid ammonia at different injection angles and injection distances. Therefore, it is innovative to ignite liquid ammonia with dimethyl ether under different injection distances, different injection angles and different liquid ammonia energy ratios.

Sincerely,

Jun Fu, PhD

4160@hnsyu.edu.cn

College of Mechanical and Energy Engineering, Shaoyang University, Shaoyang , China.

---

## [Decision Letter · Decision Letter 1]

2 Sep 2025

Study on Spray Combustion Characteristics of Liquid Ammonia/Dimethyl Ether Dual Fuel Based on Different Injection Strategies

PONE-D-25-39351R1

Dear Dr. Fu,

We’re pleased to inform you that your manuscript has been judged scientifically suitable for publication and will be formally accepted for publication once it meets all outstanding technical requirements.

Kind regards,

Krishnamoorthy Ramalingam

Academic Editor

PLOS ONE

Additional Editor Comments (optional):

Reviewer #1:

Reviewer #2:

Reviewers' comments:

Reviewer's Responses to Questions

**Comments to the Author**

Reviewer #1: All comments have been addressed

Reviewer #2: (No Response)

2. Is the manuscript technically sound, and do the data support the conclusions?

Reviewer #1: Yes

Reviewer #2: (No Response)

3. Has the statistical analysis been performed appropriately and rigorously?

Reviewer #1: Yes

Reviewer #2: (No Response)

4. Have the authors made all data underlying the findings in their manuscript fully available?

Reviewer #1: Yes

Reviewer #2: (No Response)

5. Is the manuscript presented in an intelligible fashion and written in standard English?

Reviewer #1: Yes

Reviewer #2: (No Response)

Reviewer #1: (No Response)

Reviewer #2: The authors improved the paper quality based on the reviwer's comments. The paper can be accpeted as it is.

**Do you want your identity to be public for this peer review?** For information about this choice, including consent withdrawal, please see our Privacy Policy

Reviewer #1: No

Reviewer #2: No

---

## [Editor Report · Acceptance letter]

PONE-D-25-39351R1

PLOS ONE

Dear Dr. Fu,

I'm pleased to inform you that your manuscript has been deemed suitable for publication in PLOS ONE. Congratulations! Your manuscript is now being handed over to our production team.

Kind regards,

on behalf of

Dr. Krishnamoorthy Ramalingam

Academic Editor

PLOS ONE